# A structural biology community assessment of AlphaFold2 applications

Mehmet Akdel[1,19], Douglas E. V. Pires[2,19], Eduard Porta Pardo[3,4,19], Jürgen Jänes [5,19], Arthur O. Zalevsky [6,19], Bálint Mészáros[7,19], Patrick Bryant [8,19], Lydia L. Good [9,19], Roman A. Laskowski [5,19], Gabriele Pozzati [8], Aditi Shenoy [8], Wensi Zhu[8], Petras Kundrotas[8], Victoria Ruiz Serra [4], Carlos H. M. Rodrigues [2], Alistair S. Dunham [5], David Burke [5], Neera Borkakoti[5], Sameer Velankar [5], Adam Frost [10], Jérôme Basquin[11], Kresten Lindorff-Larsen [9], Alex Bateman [5], Andrey V. Kajava [12], Alfonso Valencia [4] ✉, Sergey Ovchinnikov [13] ✉, Janani Durairaj [14] ✉, David B. Ascher [15] ✉, Janet M. Thornton [5] ✉, Norman E. Davey [16] ✉, Amelie Stein [9] ✉, Arne Elofsson [8] ✉, Tristan I. Croll [17] ✉ & Pedro Beltrao [5,18] ✉

Most proteins fold into 3D structures that determine how they function and orchestrate the biological processes of the cell. Recent developments in computational methods for protein structure predictions have reached the accuracy of experimentally determined models. Although this has been independently verified, the implementation of these methods across structural-biology applications remains to be tested. Here, we evaluate the use of AlphaFold2 (AF2) predictions in the study of characteristic structural elements; the impact of missense variants; function and ligand binding site predictions; modeling of interactions; and modeling of experimental structural data. For 11 proteomes, an average of 25% additional residues can be confidently modeled when compared with homology modeling, identifying structural features rarely seen in the Protein Data Bank. AF2-based predictions of protein disorder and complexes surpass dedicated tools, and AF2 models can be used across diverse applications equally well compared with experimentally determined structures, when the confidence metrics are critically considered. In summary, we find that these advances are likely to have a transformative impact in structural biology and broader life-science research.

Proteins are the key molecules of the cell that are involved in all cellular processes. The three-dimensional (3D) shape of a protein provides critical information that can, among many things, be used to study protein interactions, functions and the impact of missense variation. Although tremendous progress has been made in experimental approaches to determining protein structures, the experimentally determined structures of ~100,000 proteins[1] represent a very small fraction of the size and diversity of the universe of proteins. Protein structure prediction has been a fundamental challenge in bioinformatics for decades, and accurate predictions could accelerate our understanding of protein structure–function relationships, with vast impacts on the study of life. Since the first blind assessment of prediction methods, much progress has been made—including improvements in extracting pair-wise and higher order residue distance constraints from multiple sequence

A full list of affiliations appears at the end of the paper. ✉e-mail: alfonso.valencia@bsc.es; so@fas.harvard.edu; janani.durairaj@gmail.com; d.ascher@uq.edu.au; thornton@ebi.ac.uk; norman.davey@icr.ac.uk; amelie.stein@bio.ku.dk; arne@bioinfo.se; tic20@cam.ac.uk; pbeltrao@ethz.ch

alignments[2–5], and an understanding of how this information is eventually encoded into a predicted 3D structure[6–8]. These developments have been reviewed recently[9] and can be characterized by the increasing use of neural-network models in key aspects of the challenge of predicting protein structures from their primary sequence. Along with advances in computational methods, there have been large expansions of protein sequence and structure databases[10,11], which have served as key resources for input and training of sophisticated prediction methods. These advances have led to the recent leap in performance demonstrated by Deepmind at CASP14 (ref.[12]). AF2 has been shown to be able to predict the structure of protein domains with an accuracy matching that of experimental methods. Both the method and a database of 365,198 protein models have been released[13], enabling the scientific community to better understand the accomplishments, abilities and limitations of AF2.

The accuracy of AF2 has been independently evaluated in blind assessments. Yet many questions remain regarding the extent to which these approaches extend our coverage of structural biology, and the limitations of the AF2 method or structures derived from AF2 for applications in biology. Regarding coverage, previous attempts to generate 'proteome-wide' structural models include those based on homology models, such as the SWISS-MODEL Repository (SMR)[14], and more recently, the modeling of known protein domains in the Pfam database[15] using trRosetta[16]. These represent prior benchmarks of large target coverage that can be a useful comparison for AF2's performance.

With regard to the application of AF2 structures, it is noteworthy that the platform provides metrics of uncertainty[12] that have been shown to reflect confidence in the structural assignment—potentially linked to protein disorder—and uncertainty for pair-wise residue distances. It is therefore important to assess whether AF2 structures and confidence metrics can be successfully integrated into and applied to critical structural biology tasks, such as functional classification, variant effects, binding site prediction and modeling into new experimental data (obtained, for example, from cryogenic electron microscopy (cryo-EM)). In addition to the prediction of individual protein structures, it has been shown recently that contact predictions can be used to simultaneously fold and dock proteins[17], and early reports have indicated that AF2 can predict the structure of complexes[18–20], which it was not initially trained to handle.

Here, we provide an evaluation and practical examples of applications of AF2 predictions across a large number of diverse structural biology challenges.

## Results

### Added structural coverage by AlphaFold2 predictions of model proteomes

The AF2 database has released predictions of the canonical protein isoforms for 21 model species, covering nearly every residue in 365,198 proteins. This represents around twice the number of experimental structures and six times the number of unique proteins in the Protein Data Bank (PDB). It is important to assess the extent to which AF2 predictions extend the structural coverage beyond previous proteome-wide structural predictions. We compared the structures of 11 model species that were included in both the SMR and AF2 databases and that had an average additional coverage of 44% of residues by AF2 (Fig. 1a, residues). However, not all of AF2's residue predictions have high confidence. For residues that are not present in the SMR, we observed that an average of 49.4% are predicted with confidence by AF2 (predicted local distance difference test score (pLDDT) > 70) (Fig. 1a, AF residue confidence). With a more stringent cut-off (pLDDT > 90), AF2 predicts, on average, 25% of residues with very high confidence. In summary, an average of around 25% of the residues of the proteomes of the 11 model species are covered by AF2 with novel (not present in SRM) and confident (pLDDT > 70) predictions.

We then compared AF2 predictions with those derived for Pfam protein domains[15] using trRosetta[16]. As there is only one trRosetta representative structure per domain family, we selected one species—human—and compared 3,035 AF2 models of 1,464 different Pfam domain families with the representative trRosetta model. These two approaches generally agree, with around 50% of AF2 domain structures having a root-mean-square deviation (r.m.s.d.) < 2 Å from the generic trRosetta model (Supplementary Fig. 1a). We observed a correlation between the estimated accuracy of the AF2 model (pLDDT) and the r.m.s.d. from the trRosetta model (Fig. 1b and Supplementary Fig. 1b,c). For AF2 models with an r.m.s.d. below 2 Å from the trRosetta model have, more than 90% of their residues, on average, have a pLDDT above 70 (Fig. 1b). We also examined the variability of domain structure for 273 domain families with 3 or more instances in the human proteome (Supplementary Fig. 2), and observed that 70% of domain instances are within one s.d. of the mean r.m.s.d. for their domain family. Together, these results indicate that, for at least 50% of human Pfam domains, the trRosetta Pfam model was already likely to be accurate.

We assessed the confidence and length of AF2 contiguous regions that are not covered in SMR to identify regions that may correspond to novel structures of folded domains, rather than short termini or inter-domain linkers. The distribution of median confidence scores of a fragment versus fragment length shows an enrichment for high-confidence predictions with a length of 100–500 residues (Fig. 1c and Supplementary Fig. 3), consistent with the size of a typical protein domain[21]. This relation can be observed for all species, except *Staphylococcus aureus* (Supplementary Fig. 3). We identified, across the 11 species, 18,429 contiguous regions that are 'domain like' (with a length of 100–500 residues) with confident predictions (pLDDT > 70) that have no model in SMR. The human regions are provided in Supplementary Table 1.

Around half the residues in AF2 predictions of the 11 model species are of low confidence, many of which may correspond to regions without a well-defined structure in isolation. It has been shown that regions with low pLDDT are often intrinsically disordered proteins or regions (IDPs/IDRs)[13]. We benchmarked AF2-derived metrics against IUPred2 (ref.[22]), a commonly used disorder predictor (Fig. 1c), using regions annotated for order/disorder (Supplementary Table 2). In addition to using pLDDT, we tested the relative solvent accessible surface area (SASA) of each residue and smoothed versions of these metrics (Fig. 1d and Supplementary Fig. 4). pLDDT and window averages of pLDDT or SASA outperformed IUPred2, indicating that AF2's low-confidence predictions are enriched for IDRs. To facilitate the study of human IDRs, we provide these predictions for human proteins in Supplementary Dataset 1 and in ProViz[23]: http://slim.icr.ac.uk/projects/alphafold?page=alphafold_proviz_homepage.

### Characterization of structural elements in AlphaFold2's predicted models across 21 proteomes

The AF2 database is likely to contain structural elements that may not have been extensively seen in experimental structures. Owing to the presence of low-confidence regions in the AF2 proteins, we first split each prediction into smaller high-confidence units (see Methods). We then performed a global comparison of structural elements between the 365,198 proteins in the AF2 database and 104,323 proteins from the CASP12 dataset in the PDB. We applied the Geometricus algorithm[24] to obtain a description of protein structures as a collection of discrete and comparable shape-mers, analogous to *k*-mers in protein sequences. We then obtained a matrix of such shape-mer counts for all proteins, which we clustered using non-negative matrix factorization (NMF) (see Methods). The clustering identified 250 groups of proteins, dubbed 'topics' (Supplementary Dataset 2), with characteristic combinations of shape-mers. These characteristic shape-mers could include small structural elements, such as repeats, the specific arrangements of ion-binding sites or larger structural elements that could define specific folds. For visualization, we performed a *t*-distributed stochastic

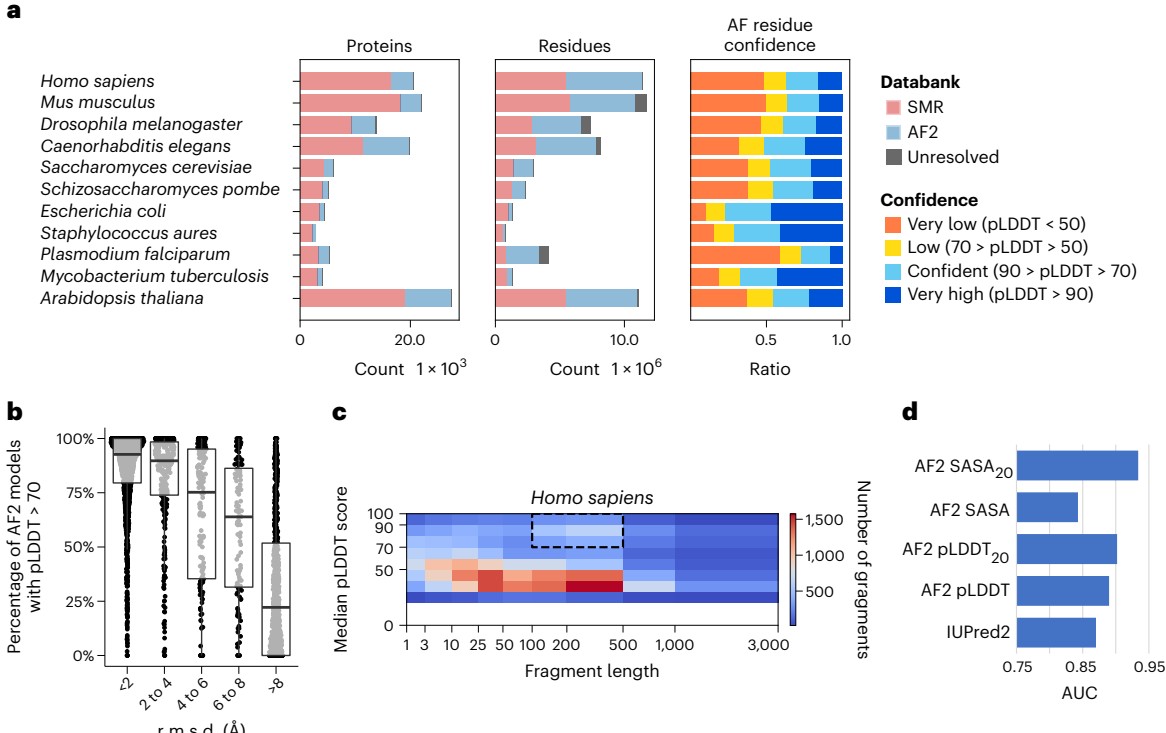

**Fig. 1 | Additional coverage provided by AF2-predicted models. a**, Added structural coverage (per-protein, left; per-residue, middle) and per-residue confidence of regions not covered by SMR (right) for 11 species included in both the AF2 and SMR databases. **b**, Fraction of confident (pLDDT > 70) residues per human AF2 model, binned by r.m.s.d. from the corresponding trRosetta-derived domain-level Pfam model; 3,035 AF2 predicted structures of protein regions matching one of 1,464 different Pfam domain families were compared with the corresponding trRosetta model. **c**, Median fragment length and median pLDDT score of human AF2-only regions. The highlighted area identifies high-confidence regions with domain-like length. The bottom, middle line and top of the box correspond to the 25th, 50th and 75th percentiles, respectively. **d**, Comparison of AF2 SASA (SASA$_{20}$, 20-residue smoothing) and pLDDT (pLDDT$_{20}$, 20-residue smoothing) against a disorder prediction method (IUpred2).

neighbor embedding (*t*-SNE) dimensionality reduction in which proteins composed of similar shape-mers are expected to group together (Fig. 2). In line with this, the shape-mer representation of AF2 proteins can predict the corresponding PDB protein entries with high accuracy (area under the receiver operating characteristic curve of 0.95 using the cosine similarity of the shape-mer vector). Additionally, the 20 most common superfamilies, predicted from sequence, tend to be placed together.

Out of 250 total groups, we selected 5 examples that were almost exclusively (>90%) composed of structures derived from AF2, as well as 1 example with >80% AF2 structures with a particularly interesting novel predicted structural element. We illustrated these with a representative structure in Figure 2. Examples include 4,192 proteins annotated as G-protein-coupled olfactory or odorant receptors (Pfam PF13853), 97% of which are mammalian (Fig. 2a, Topic 88, and Supplementary Fig. 5a); a group of primarily (94%) plant proteins, annotated as PCMP-H and PCMP-E subfamilies of the pentatricopeptide repeat (PPR) superfamily (Fig. 2b, Topic 60, and Supplementary Fig. 5b); a group of heterogeneous structures that were mostly (>75%) annotated as ATP or ion binding (Fig. 2c, Topic 150, and Supplementary Fig. 5c); groups of proteins with leucine-rich repeats (Fig. 2d, Topic 16, and Supplementary Fig. 5d); some proteins with uncommon, regular patterns (Fig. 2e, Topic 188, and Supplementary Fig. 5e); and long α-helical constructs (Fig. 2f, Topic Helix, Supplementary Fig. 5f). For the PCMP-H and PCMP-E subfamilies (Fig. 2b), there are no known experimental structures mapped. AF2 predictions could help elucidate the structural peculiarities of these subfamilies, including the mechanism of RNA recognition and binding for PCMP-H and PCMP-E proteins.

Studying examples from *Mycobacterium tuberculosis* in Topic 188 led us to identify an interesting structure for a tandem repeat. Tandem repeat proteins with repetitive units of 6–10 residues predominantly have beta-solenoid structures[25]. Analyzing the AF2 results, we found a novel beta-solenoid structure predicted for a large family of pentapeptide repeats[26], found in the mycobacterial PPE proteins (Pfam: PF01469) (Fig. 2e and Supplementary Fig. 6). This structure represents a beta-solenoid, with the shortest possible coil of ten residues (two pentapeptide repeats) (Supplementary Fig. 6b). Although such a beta-solenoid has not yet been resolved, our evaluation of the quality of the atomic structure (stereochemistry and contacts) suggests that the AF2 model is highly probable. Thus, AF2 may have allowed us to answer the question of what is the shortest length of repeat that forms a beta-solenoid.

Finally, we also considered protein groups consisting primarily of PDB proteins to study why AF2 proteins are absent from them. In some cases, this seemed to be due to the limited number of species and proteins covered by the current AF2 database. Topics 209 and 113 consist of immune response proteins, such as immunoglobulins and T-cell receptors, mainly from the PDB. As many of these antibodies are under intense study, there are many more PDB structures (based on multiple individuals and antibody-drug research) than the actual number of such proteins in the respective UniProt proteomes. Topic 38 consists of short fragments of PDB structures, with an average length of 63 residues—there are no AF2 proteins, because AlphaFold models the entire structure instead of returning fragments.

## Application of AlphaFold2 models for structure-based variant effect prediction

A protein structure facilitates the generation of hypotheses regarding the impact of missense mutations. Conversely, an agreement between the expected and observed impacts of mutations provides confidence

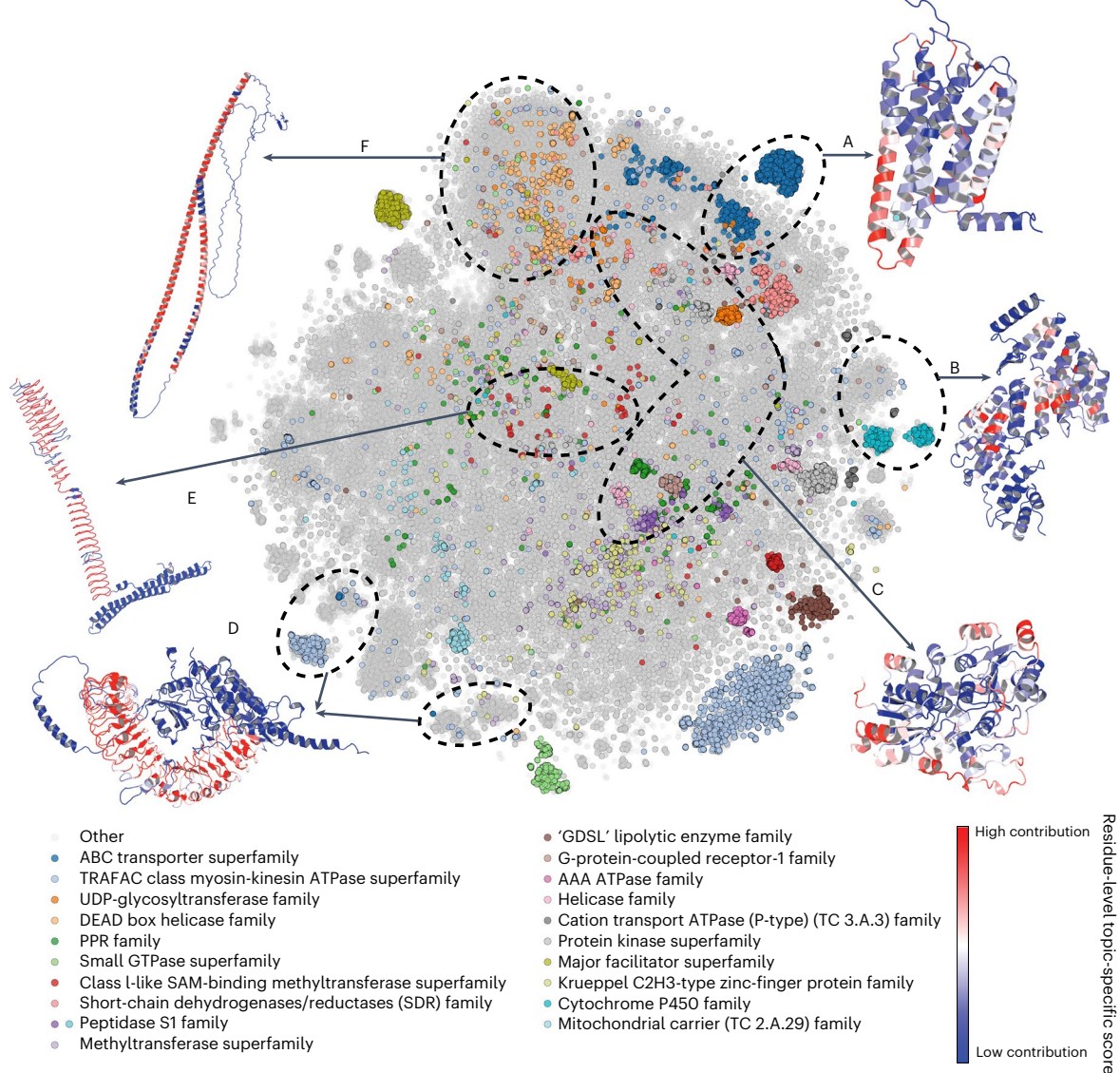

**Fig. 2 | The space of characteristic structural elements in AF2 structural models for 21 species.** Visualization of *t*-SNE dimensionality reduction analysis, in which structures with similar structural elements are placed closer together and the 20 most common superfamilies are colored. The axes corresponding to the *t*-SNE dimension 1 and *t*-SNE dimension 2 were omitted. Six shape-mer groups (that is, topics) discussed in the text, consisting of mainly AF2 proteins

as opposed to PDB proteins, are labeled A–F, and a representative structure is depicted for each. Residues in the representative structures are colored according to their contribution to the topic under consideration—red residues have the highest contribution, and blue residues are specific to the example and not to the topic.

in the accuracy of a structural model. We obtained two independent compilations of experimentally measured impacts of protein mutations on protein function: (1) a compilation of measured changes in stability upon mutations[27,28]; and (2) a compilation of deep mutational scanning (DMS) experiments[29,30] measuring the outcome of any possible single point mutation on most protein positions.

The DMS data were available for 33 proteins with 117,135 mutations; we obtained experimentally derived models for 31 of the proteins and AF2 models for all 33. We then used three structure-based variant effect predictors (FoldX[31], Rosetta[32] and DynaMut2 (ref. [33])) to compare the DMS measurements with predicted impacts. Although the correlation estimates between the experimental and predicted impacts of mutations varied across the proteins, those derived from the AF2 models consistently matched or were better than those derived from experimental models (Fig. 3a,b and Supplementary Fig. 7). Regions with confidence scores lower than 50 result in lower concordance

(Fig. 3a), but restriction to protein regions without an experimental model can still lead to correlations that are comparable to those observed in experimental structures (Fig. 3b). Because low AF2 confidence scores are enriched for intrinsically disordered protein regions, it is possible that the poor correlation in low-confidence regions is in part owing to higher tolerance to protein mutations. In line with this, we observed an average higher tolerance to mutations in low-confidence regions (Fig. 3c).

The compilation of measured impacts of mutations on protein stability contains information for 2,648 single-point missense mutations over 121 distinct proteins. We compared the accuracy of structure-based prediction of stability changes using AF2 structures, experimental structures and homology models using different sequence identify cut-offs (Fig. 3d and Supplementary Fig. 8; see Methods). Across 11 well-established methods (Fig. 3d and Supplementary Fig. 8), the predictions of stability changes based on AF2 models were comparable to

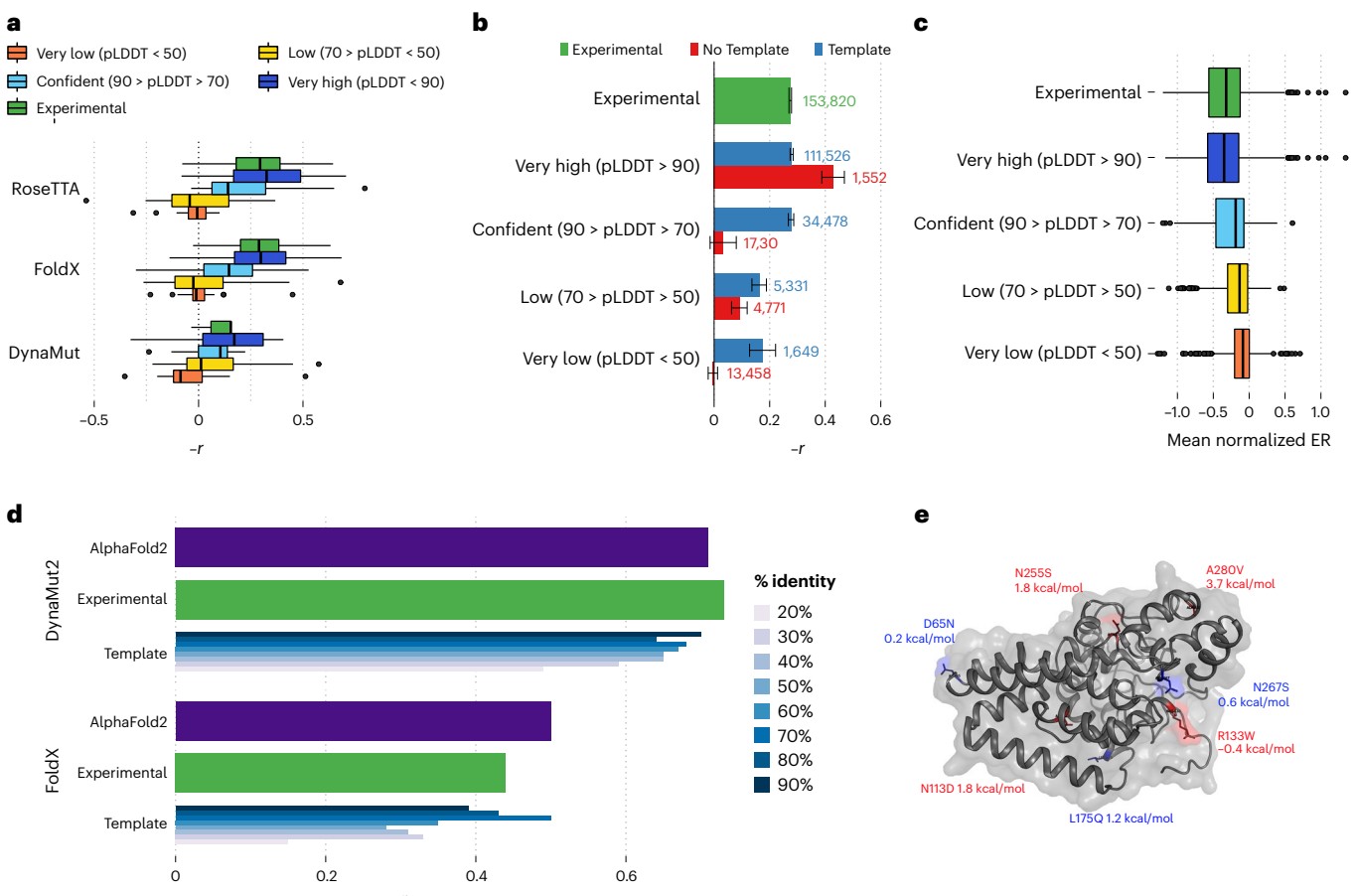

**Fig. 3 | Comparing structure-based prediction of impact of protein missense mutations using experimental and AF2-derived models. a**, Relationship between the predicted $\Delta\Delta G$ for mutations with measured experimental impact of the mutation from deep mutational scanning data ($-1 \times$ Pearson correlation). The predicted change in stability was determined using one of three structure-based methods, using structures from AF2 or available experimental models. The bottom, middle line and top of the box correspond to the 25th, 50th and 75th percentiles, respectively. The lines extend to $1.5 \times$ IQR (interquartile range). A total of 117,135 mutations were used in the analysis. **b**, Correlations based on the FoldX predictions as in **a**, but subsetting the positions in AF2 models according to confidence and whether the position is present in an experimental structure. Data are presented as mean values ± the confidence intervals calculated via fisher's $Z$ transform (R's cor.test function). **c**, The mean impact of a mutation,

calculated as the enrichment ratio (ER) score, from DMS data for positions in AF2 models with different degrees of confidence. A total of 117,135 mutations were used in the analysis. **d**, Comparative performance of methods for predicting stability changes upon mutation using AF2 and experimental and homology models based on protein structure templates of different identity cut-offs. Experimental measurements of stability are for 2,648 single-point missense mutations over 121 proteins. The bottom, middle line and top of the box correspond to the 25th, 50th and 75th percentiles, respectively. The lines extend to $1.5 \times$ IQR. **e**, Example application for structure-based prediction of stability impact of known disease mutations for a human protein with little structural coverage prior to AF2. $\Delta\Delta G$ stability changes were predicted using Rosetta, and a substantial impact was considered for $\Delta\Delta G > 1.5$ kcal/mol.

those of experimental structures. Homology-model-based predictions tended to show substantial decreases in performance for templates below 40% sequence identity.

We investigated, as an example, the human Sphingolipid delta(4)-desaturase (DEGS1), a 323-residue protein associated with leukodystrophy, for which no structure or model was available. All but the terminal residues are predicted by AF2 with high confidence. The presumed catalytic core is discussed further below. Here we focus on disease-associated missense variants. p.A280V has been shown to lead to loss of protein stability[34] and has a predicted Gibbs free energy change ($\Delta\Delta G$) of 3.7 kcal/mol. Two additional pathogenic variants have $\Delta\Delta G$ values of >1.5 kcal/mol, pointing towards loss of stability being the mechanism of pathogenicity; the benign variants do not substantially affect protein stability, as expected (Fig. 3e). The likely pathogenic variant p.R133W is not predicted to affect stability, and hence likely has a different mechanism underlying disease. This is in line with previous findings that core variant changes in particular

lead to loss of stability, whereas surface variants are more likely to act through other mechanisms[30].

## Functional characterization of AF2 models by pocket and structural motif prediction

High-confidence proteome-wide structural predictions open the door for a large expansion of predicted protein pockets[35,36]. However, the full protein models produced by AF2 have to be considered carefully given their potential errors, such as the likely incorrect placement of protein segments of low confidence or the low confidence in interdomain orientations. To investigate whether these issues may result in the formation of spurious pockets, we predicted pockets on a set of 225 proteins with known binding sites defined using bound (holo) structures for which the corresponding unbound (apo) structures are available[37].

Pockets identified from structures have a wider size range than do ground-truth binding sites (Fig. 4a). This is also true for pockets predicted from AF2 structures, including a small number of

particularly large pockets (Fig. 4a). We divided AF2 pocket predictions into high-quality (mean pLDDT > 90) and low-quality (mean pLDDT ≤ 90) subsets (Fig. 4b,c) on the basis of the mean pLDDT of pocket-associated residues. Low-quality pockets are larger on average, and include particularly large pockets (Fig. 4a, bottom). We then asked whether mean pLDDT could be useful as a general metric of prediction confidence by quantifying the overlap between known and predicted pockets (Fig. 4b and Supplementary Fig. 9). We did not observe a difference between the performance of high-quality AF2 pockets and pockets identified from experimental structures. In contrast, low-confidence pockets generally did not overlap with known sites. Although there may be bias because high-confidence AF2 regions are more likely to have relevant deposited templates, we suggest that the mean pLDDT of predicted pockets can be used as an additional criterion for pocket selection in AF2 structures.

Conserved local conformations of specific residues can be used to identify important functions, such as enzyme activity, ion or ligand binding beyond global sequence and fold similarities[38]. To showcase the potential of this application for AF2 models in the future, we focused on 912 human proteins with no experimental or homology models available. We found that the prediction score of the highest ranked pocket enriched the set for proteins with previous annotations for enzymatic activity (Fig. 4c and Supplementary Table 3). Discarding pockets with a low mean pLDDT led to slightly improved enrichment. As a specific example, we focused on the human sphingolipid delta(4)-desaturase (EC 1.14.19.17, DEGS1, UniProt Accession O15121, pocket score rank 57 of 912), which has a high confidence level (average pLDDT = 96.31) and for which there are no previous structural data. A sequence search of the 323-residue protein against all existing entries in the PDB shows that the best sequence match is 23.5%, with PDB entry 1VHB (Bacterial dimeric hemoglobin, 9115439), indicating the lack of any structural models from homology. A scan of 400 auto-generated 3-residue templates from the AF2-predicted structure against representative structures in the PDB (reverse template comparison[38]) yielded a possible 3-residue template match: PDB entry 4ZYO (EC 1.14.19.1, human stearoyl-CoA desaturase[39], Fig. 4d). A close up of the metal-binding center (Fig. 4e) of DEGS1 and 4YZO (overall sequence homology, 12.1%) superimposed via the 3-residue templates (Fig. 4d) clearly indicates the potential dimetal catalytic center for DEGS1. The histidine-coordinating metal center of DEGS1, together with data on the bound substrate of 4ZYO, provides a foundation for modeling studies that could impact the pharmacology of DEGS1 by exploring the details of its catalytic mechanism.

## AlphaFold2-based prediction of protein complex structures

Since the first development of direct coupling analysis algorithms, co-evolutionary-information-based methods have been used to predict protein–protein interactions[40]. It has been recently reported that several deep-learning-based methods, such as trRosetta[16] and Raptor-X[41], can predict the structure of protein complexes. To examine the capacity of AF2 to predict protein complex structures, we tested the ability of AF2 to fold and 'dock' two benchmark sets—a set of proteins known to form oligomers[42] and the Dockground 4.3 heterodimeric benchmark[43].

For oligomerization, we obtained sets of proteins known either not to oligomerize or to form oligomers, including dimers, trimers or tetramers. We then made AF2 predictions for each protein, attempting to predict either a monomer or an oligomeric form (see Methods). Across the set of predictions, higher scores were given to models corresponding to the correct oligomerization state, and 71 out of 87 (82%) predicted top-scoring models corresponded to the correct state (Fig. 5a and Supplementary Table 4). Generally, the multimeric state scores are well separated from the monomeric state scores (Fig. 5b). In 28/30 examples, AF2 was able to correctly predict monomeric proteins as monomers, 29/35 dimers as dimers, 7/9 trimers as trimers and 7/13 tetramers as tetramers. Notably, although the failure rate is high for tetramer state predictions, the predicted structure for the

corresponding state was actually correct for 5/6 failures. Examples of failure modes for dimers and a tetramer are shown in Figure 5c,d. We noted that, for some cases of failed tetramer predictions, we could obtain higher confidence of the tetramer predictions by increasing the number of recycles.

We next examined the Dockground 4.3 heterodimeric benchmark set[43]. We predicted complex structures using the DeepMind default dataset and the small Big Fantastic Database (BFD) database. This method does not include any 'pairing' of interacting chains, as was used in earlier fold-and-dock approaches. The docking quality was evaluated using DockQ[44,45]. Only one model for each target was made, and a maximum of three recycles were allowed. In Figure 5e, it can be seen that the performance is far superior to traditional docking methods, with 31% of correctly predicted protein complex models, compared with 7% using GRAMM, a standard shape-complementarity docking method[44].

Finally, we studied examples of complexes containing IDPs/IDRs that adopt a stable structure upon binding. IDRs often bind through short linear motifs (SLiMs), recognizing folded domains driven by a few residues. The longer IDRs can contain arrays of SLiMs and can also form stable structures upon binding to other IDRs without a structured template. We selected 14 cases of complexes involving IDRs with known structures and analyzed their distinguishing features compared with the experimental complex (Fig. 5f contains selected examples and Supplementary Figs. 10 and 11 show all examples). In general, AF2 performs well at predicting SLiMs that fit into a well-defined binding pocket driven by hydrophobic interactions, such as the SUMO interacting motif of RanBP2. Longer IDRs, which frequently contain tandem motifs, are often challenging, especially if they have a symmetric structure. For the RelA–CBP interaction, AF2 correctly finds the binding groove, but fits the IDR in a reverse orientation. AF2 also performs well on complexes in which IDRs are part of a multi-IDR single folding unit, such as the E2F1–DP1–Rb trimer; however, building complexes for proteins with highly unusual residue compositions, such as collagen triple helices, often fail. We provide a detailed description of the 14 examples in Supplementary Figures 10 and 11 and Supplementary Table 5 and detail the factors that enable or hinder successful predictions.

## Evaluation of AlphaFold2 models for use in experimental model building

The accuracy of AF2 predictions provides opportunities for their use in experimental model building: (1) AF2 models could be used for molecular replacement or docking into cryo-EM density, experimental phasing and/or ab initio model building; and (2) they could be used as reference points to improve existing low-resolution structures. These use cases will typically involve the use of conformational restraints, for example to maintain the local geometry of domains while flexibly fitting a large multi-domain model, or to restrain the local geometry of an existing model of an AF2-derived reference to highlight and correct likely sites of error. It is critical to use restraint schemes designed to avoid forcing the model into conformations that clearly disagree with the data. Typically, this is achieved through some form of top-out restraint, for which the applied bias drops off at large deviations from the target. Here, we take advantage of the fact that AF2 models typically include very strong predictions of their own local uncertainty to adjust per-restraint weighting of the adaptive restraints recently implemented in ISOLDE[46] (see Methods). For the two case studies discussed below, a comparison of validation statistics for the original and revised models is provided in Supplementary Table 6.

As an example of the improvement of existing structures, we used the eukaryotic translation initiation factor (eIF) 2B bound to substrate eIF2 (6O85)[47,48]. The eIF2B complex is a decamer comprising two copies each of five unique chains. It displays allosteric communication between physically distant substrate-, ligand- and inhibitor-binding sites. eIF2 is a heterotrimer of three unique chains. We analyzed a 0.4-MDa co-complex enzyme-active state captured by cryo-EM at an

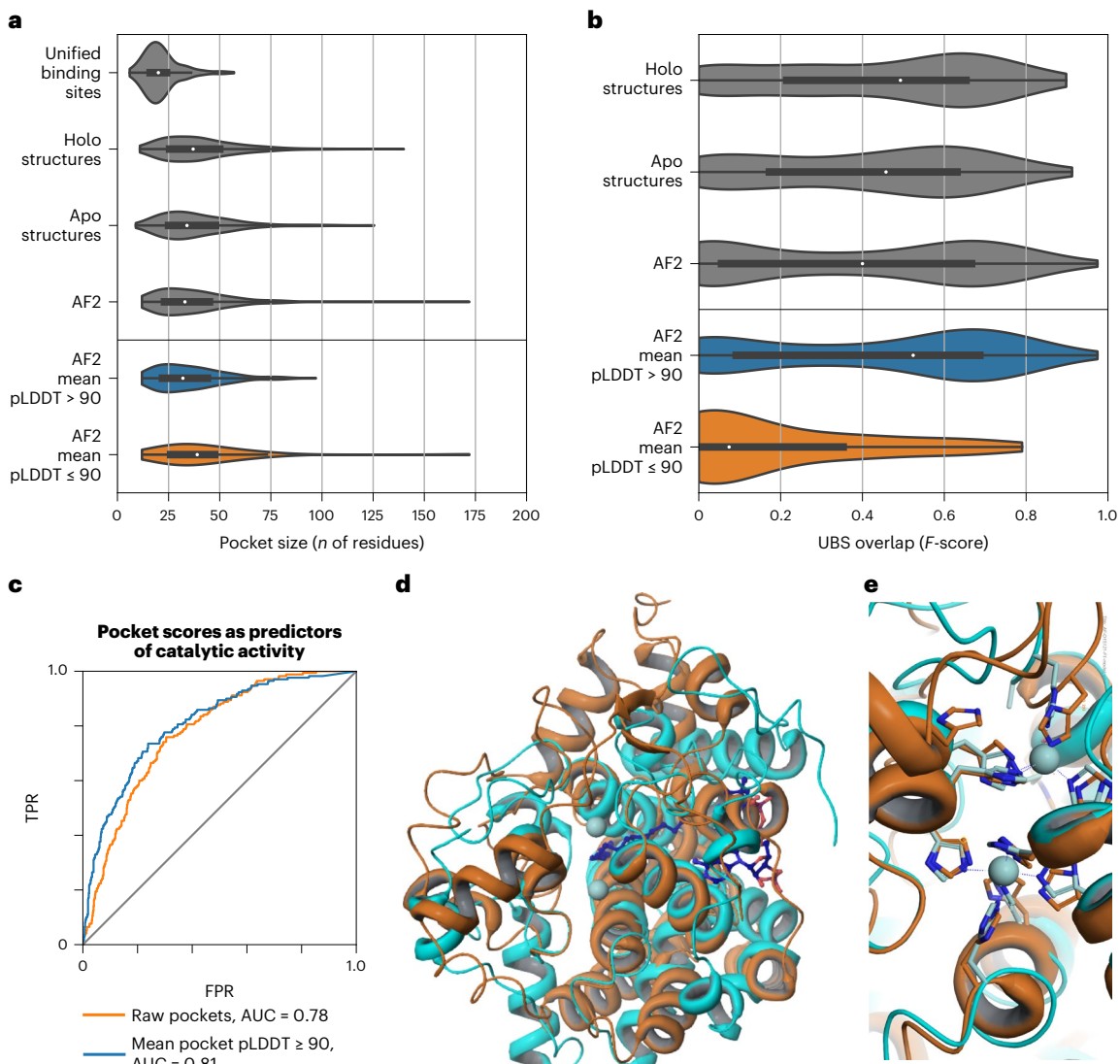

**Fig. 4 | Pocket detection and function prediction. a**, Size of known binding sites (or unified binding sites) compared with the size of top AutoSite pockets in experimental holo (bound), experimental apo (unbound) and AF2 structures. AF2 structures are split into high-confidence (mean pLDDT > 90) and low-confidence (mean pLDDT ≤ 90) subsets. The bottom, middle line and top of the box correspond to the 25th, 50th and 75th percentiles, respectively. The lines extend to 1.5 × IQR. **b**, Distribution of overlap between known binding sites and top predicted pockets for holo, apo and AF2 structures. The bottom, middle line and top of the box correspond to the 25th, 50th and 75th percentiles, respectively. The lines extend to 1.5 × IQR (interquartile range). **c**, Enzymatic activity prediction using pocket-derived, template-derived and combined metrics. AUC, area under the curve. TPR, true positive rate; FPR, false positive rate. **d**, Superposition of the AF2 model of DEGS1 (O15121) with PDB entry 4ZYO. Orange: ribbon representation of AF2 predicted structure for DEGS1. Cyan: ribbon representation of 4ZYO. Zinc atoms (light blue spheres) and bound substrate (dark blue ball and stick) as observed in the structure of 4ZYO are also shown. **e**, Close up of the metal-binding center of 4ZYO. Ribbon representation of the protein and metal chelators for DEGS1 and 4ZYO are shown in orange and cyan, respectively. The zinc atoms observed in 4ZYO are shown as light blue spheres. Metal-chelating residues for DEGS1 are clearly identifiable.

overall resolution of 3 Å (ref. [49]). Rigid-body alignment of AF2 models to their corresponding experimental chains (Fig. 6a) showed overall excellent agreement, with the largest deviations corresponding to correctly folded domains with flexible connections to their neighbors. Other mismatched smaller regions corresponded to either register errors in the original model or flexible loops and tails. Each chain was restrained to its corresponding AF2 model using ISOLDE's reference-model distance and torsion restraints, with each distance restraint adjusted according to pLDDT. Future work will explore the use of the predicted aligned error (PAE) matrix for this purpose, and weighing of torsion restraints according to pLDDT. Simple energy minimization and equilibration of the restrained model at 20 K corrected the majority of local geometry issues (for example, Fig. 6b,c); a high-confidence prediction for the C-terminal domain of chains I and J allowed us to add this into

previously untraceable low-resolution density (Fig. 6d, left of the dashed line). We emphasize that detailed manual inspection remains necessary to find and correct larger errors in the experimental model, sites of disagreement arising from conformational variability and sites where high-confidence predictions are in fact incorrect. An example of the latter is the side chain of Trp A111, which, despite its high confidence (pLDDT = 86.1), was modeled incorrectly by AF2 (Fig. 6f).

To explore the use of AF2 structures for solving and refining new structures, and to map out suitable workflows, we attempted to recapitulate the recent 3.3-Å crystal structure of the *Saccharomyces cerevisiae* Nse5/6 complex (7OGG)[50]. This was not included in the AF2 training set, and no existing structures have ≥30% identity to either chain. Originally solved using selenomethionine experimental phasing, the combination of low-resolution and anisotropy ($\Delta B = 80$ Å$^2$) meant

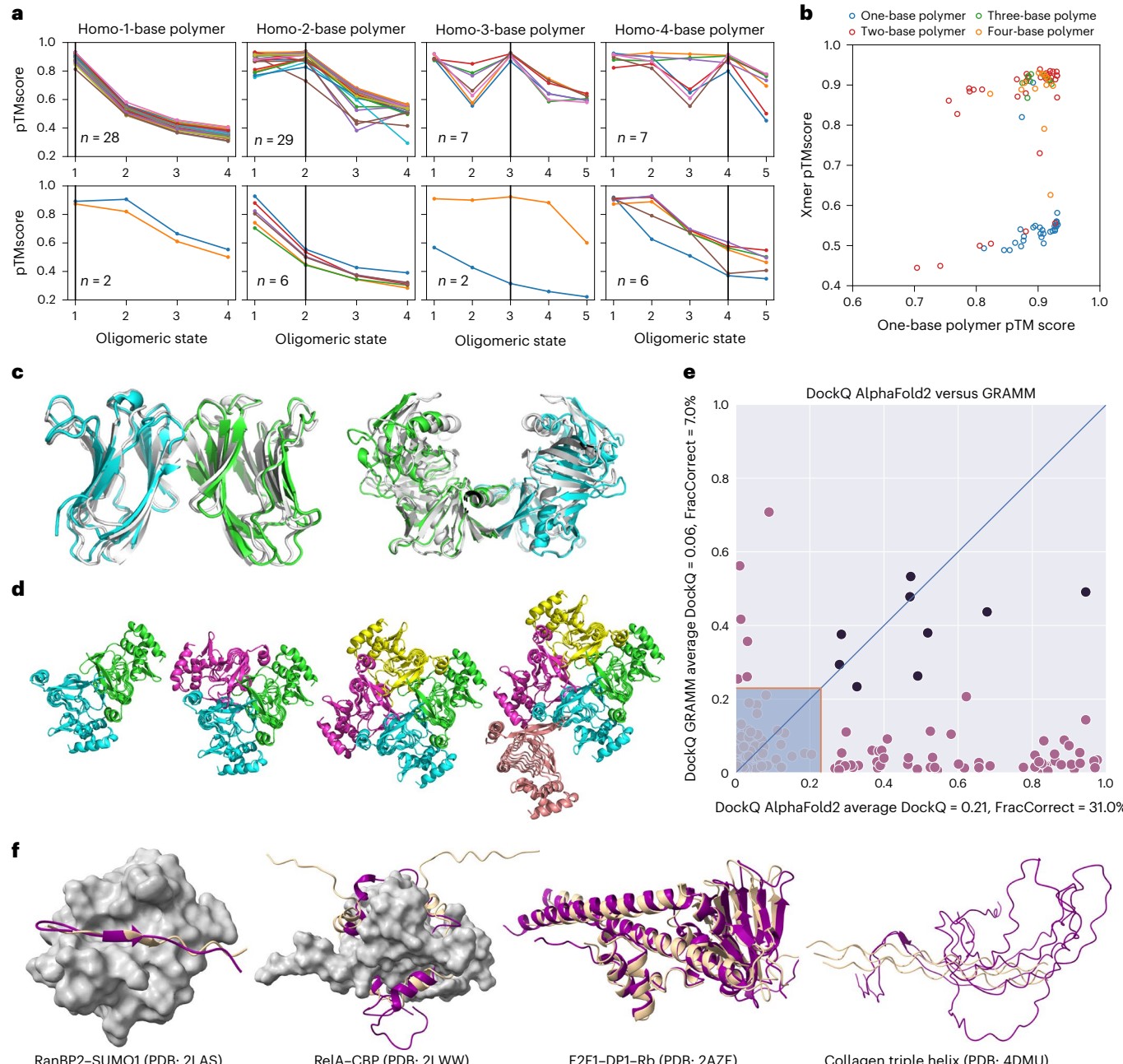

**Fig. 5 | Using AF2 to predict homo-oligomeric assemblies and their oligomeric state. a**, AF2 prediction for each oligomeric state (1–4 for monomers and dimers, and 1–5 for trimers and tetramers). Only proteins for which the monomer had pLDDT > 90 are shown. For visualization, the predicted successes (top) and failures (bottom) were separated into two plots. Success is defined when the peak of the homo-oligomeric state scan matches the annotation, or the pTMscore of the next oligomer state is substantially lower (−0.1). **b**, For each of the annotated assemblies, the pTMscore of monomeric prediction is compared with the max pTMscore of non-monomeric prediction. **c**, Monomer prediction failure. Two monomers were predicted to be homo-dimers. For the first case (PDB: 1BKZ), the prediction matched the asymmetric unit (shown as blue/green and prediction in white). For the second case (PDB: 1BWZ), the prediction matched one of the crystallographic interfaces. **d**, 3TDT trimer was predicted to be a tetramer. Although the interface is technically correct, for this c-symmetric protein, the pTMscore was not able to discriminate between 3 and 4 copies. **e**, Comparison of docking quality between AF2 (x axis) and a standard docking tool GRAMM (y axis). Comparisons were made using the DockQ score. Models with a DockQ score that was higher than 0.23 are assumed to be acceptable according to the Critical Assessment for Predicted Interactions (CAPRI) criteria (marked outside the shaded area). Black circles indicate the complex was well modeled by both methods. The average DockQ score and the number of acceptable or better models are shown in the axis labels. It should be noted here that AF2 both folds and docks the proteins, whereas GRAMM only docks them. **f**, Examples of AF2-predicted interactions mediated by regions of intrinsic disorder.

that, although the core of the complex was confidently and correctly modeled, only 583 out of 850 total residues were definitively modeled by the authors, with a further 65 residues traced as unknown sequence and one peripheral 27-residue helix modeled out of register. For testing purposes, we discarded this model and used the AF2 predictions for

molecular replacement (MR). MR requires very close correspondence between atom positions in the search model and in the crystal; separation into individual rigid domains and trimming of flexible loops is a necessity. We used the PAE matrix to extract a single rigid core from each chain (see Methods) and performed MR in Phaser[51], leading to

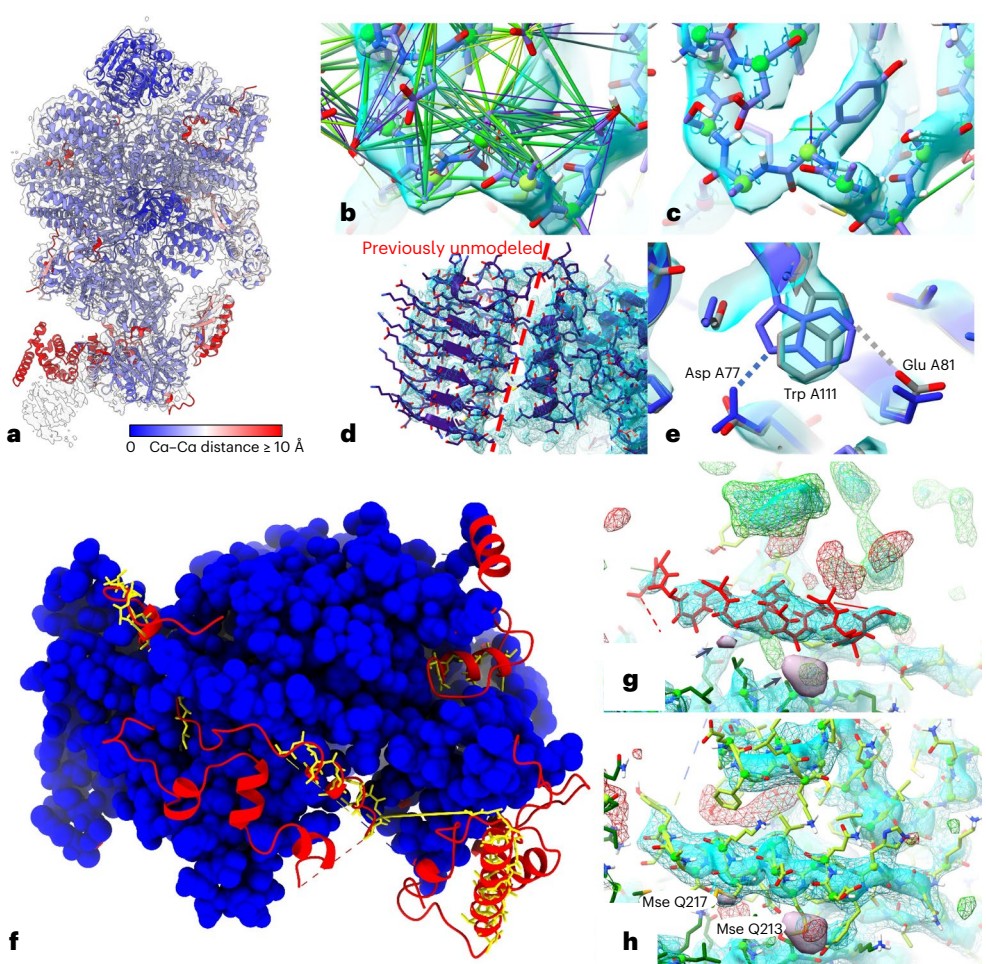

**Fig. 6 | Application of AF2 predictions to modeling into cryo-EM or crystallographic data. a**, AF2 predictions for individual chains in 6O85, aligned to the original model and colored by Cα–Cα distance, with the map (EMD-0651) contoured at 6.5 σ. Red domains at the bottom were correctly folded but misplaced owing to flexibility; smaller regions of red correspond either to flexible tails or register errors in the original model. **b,c**, Use of adaptive distance and torsion restraints to correct problematic geometry in the original model. The models before (**b**) and after (**c**) refitting are shown; satisfied distance restraints are hidden for clarity. **d**, Owing to very poor local resolution and lack of homologs, the carboxy-terminal domain in chain J (left of the dashed line) was previously left unmodeled. This domain was predicted with high confidence by AF2 (mean pLDDT = 83.0), and fit readily into the available density. **e**, High-confidence regions may still contain subtle errors that are difficult or impossible to detect in the absence of experimental data. The side chain of Trp A111 (pLDDT = 86.1) was modeled backwards (blue), forming an H-bond with Asp A77; the final model fitted to the map (gray) instead forms an H-bond with Glu A81. **f**, Rebuilding the recent 3.3-Å crystal structure 7OGG, starting from molecular replacement with AF2 models, dramatically improved model completeness. Blue, residues identified in original model; yellow sticks, residues modeled as unknown in the original model; red, residues identified in rebuilt model. **g**, Helix modeled as unknown (residues 558–573 of chain R, red), surrounded by unmodeled density (3 σ mFo-DFc, green(+), red(−); +2 σ sharpened 2mFo-DFc, cyan surface; +1.5 σ unsharpened 2mFo-DFc difference map (Fo and Fc are the experimentally measured and model-based amplitudes, D is the Sigma-A weighting factor and m is the figure of merit), cyan wireframe; +5 σ anomalous difference map, purple surface and arrows). **h**, Final model, with anomalous difference blobs corresponding to selenomethionine residues 213 and 217 of chain Q and with the previously unmodeled density filled; this region was predicted with an average pLDDT of 88, and required only minor side chain corrections to fit the density.

a clear solution with translation function Z-score (TFZ) = 28.2 and log-likelihood gain (LLG) = 884 (see Methods).

Currently, a refined MR solution is typically used as the starting point for some combination of automatic and manual building of missing portions into the density. In many cases, however, it appears that AF2 predictions will support a more 'top-down' approach, in which all residues predicted with at least moderate confidence are present in the initial model. To explore this, we trimmed the predicted chains to exclude residues with pLDDT ≤ 50 and aligned the result to the MR solution, setting the occupancies of all atoms not used for MR to zero. This was used as the starting point for rebuilding in ISOLDE; here, zero-occupancy atoms do not contribute to structure factor calculations or bulk solvent masking, but still take part in molecular interactions and are attracted into the map. The model was subjected to three rounds of end-to-end inspection and rebuilding interspersed with

refinement with phenix.refine[52]. In the initial round, zero-occupancy residues fitting the map were reinstated to full occupancy, and residues that seemed to be truly unresolved were deleted; a small number of these were re-introduced in subsequent rounds. The total time spent was approximately one working day; the final model (Fig. 6f–h) increased the number of modeled, identified residues from 600 to 818, slightly improved overall geometry and reduced the $R_{free}$ from 0.317 to 0.295. With few exceptions (primarily at heterodimer and symmetry interfaces), rebuilding was limited to minor side chain adjustments.

## Discussion

We estimate that AF2 may add, on average, around 25% of confidently predicted residues to a given proteome, although this will vary depending on how much experimental and previous computational approaches have already covered. However, even for residues that can be modeled

by distant homology, it is possible that the AF2 models are more accurate, increasing their usefulness. Here, the precise accuracy estimates at the residue level are extremely useful. In addition, low AF2 prediction scores are enriched for protein disorder, suggesting that regions of low-confidence predictions can be hypothesized to be disordered segments. However, we note that the protein disordered regions are often defined as regions that are not solved by X-ray crystallography. As AF2 is trained primarily on X-ray data, the relation between disorder and predicted confidence could be a by-product of using this definition of disorder. For comparison, we used IUPred2, an easy to deploy tool that is a commonly used dedicated protein-disorder prediction, but there are other dedicated approaches that outperform IUPred2 (ref. [53]).

The AlphaFold database was initially released with >300,000 proteins modeled with a more recent expansion to over 200 million proteins with predicted structures, sampling the universe of protein sequences and structures. As we show here, even on a relatively modest set, we can identify what are likely to correspond to rarely explored combinations of structural elements. As an example, we have identified to our knowledge the shortest length of repeat that forms a beta-solenoid to date. Among other areas, the expansion of high-confidence predictions will allow prioritization of experimental structure determination of novel folds; the large-scale prediction of protein function from structure; the identification of novel enzymes; and the study of the evolution of protein structure and function.

We assessed the application of AF2 predictions in diverse structural biology challenges, including variant effect prediction, pocket detection and model building into experimental data. In line with the reported high accuracy of the models, we found that AF2-predicted structures, on average, tend to give results that are as good as those derived from experimental structures. However, Although AF2 returns full protein predictions, these can often contain protein segments that are placed with uncertainty. This uncertainty can lead to incorrect estimations or identification of structural similarity, pockets, variant effects or poor model building. Importantly, in all cases, we find that it is critical to take into account the confidence metrics provided, and that these should be incorporated into the corresponding workflows. For model building based on experimental data, we noted examples of cases for which details were incorrect in regions where AF2 has high confidence, which underlies the need for detailed manual inspection. AF2 will not do away with experimental studies, but the combination of experimental data collection plus artificial intelligence is likely to be a growing trend.

For variant effect prediction, AF2 was used to predict the structure, and the impacts of mutations were predicted with other tools that can use these or experimental structures. A different approach could have been to use AF2 to predict the structure of the reference and mutated proteins and to compare these structures to evaluate the impact of the mutations. However, some reports have indicated that AF2 does not appear to be well suited to predict the structures of mutated proteins[54,55]. Additionally, the prediction of such large numbers of mutated structures would have been computationally time consuming.

Finally, we explored the application of AlphaFold2 for prediction of complex structures and found that it outcompetes standard docking approaches while not requiring even starting protein structures. We have expanded on this analysis in a companion manuscript[20] and have used this approach to predict complexes of human protein interactions on a large scale[56]. It has already been shown that other distance- and contact-prediction methods trained to predict unmodified intra-chain contacts can be used to predict inter-chain contact predictions, both for homo- and hetero-meric complexes. Therefore, we were not surprised to see that it is possible to use AF2 to fold and dock heterodimeric complexes. However, it was unexpected that it was possible to use non-matched pairs of alignments for different proteins to predict complex structures, indicating that AF2 goes beyond using coevolution to predict these structures.

There are many areas for further benchmarking and improvements of AF2-related approaches. Although the capacity of AF2 to predict the structures of difficult targets has been demonstrated, the extent to which AF2 generalizes to truly never before seen folds remains to be understood. The generation of a test set with folds that are not represented in the AF2 training set may not be a trivial task. Other open areas for this field of research include the prediction of: structures of mutated proteins[55]; conformation diversity and dynamics; structure in the absence of a MSA[57]; and structures of proteins in complex with other biomolecules such as DNA, RNA or metabolites. Another such area is the explicit modeling of biophysical parameters.

In summary, we find that AF2 models, when their uncertainty is taken into account, can be applied to existing structural biology challenges, and their quality is near that of experimental models. The application of AF2 to a large representation of the protein universe and expansion to the prediction of protein complexes will have a transformative impact in life sciences.

## Online content

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

[1]Bioinformatics Group, Department of Plant Sciences, Wageningen University and Research, Wageningen, the Netherlands. [2]School of Computing and Information Systems, University of Melbourne, Melbourne, Victoria, Australia. [3]Josep Carreras Leukaemia Research Institute (IJC), Badalona, Spain. [4]Barcelona Supercomputing Center (BSC), Barcelona, Spain. [5]European Molecular Biology Laboratory, European Bioinformatics Institute (EMBL-EBI), Cambridge, UK. [6]Shemyakin–Ovchinnikov Institute of Bioorganic Chemistry, Russian Academy of Sciences, Moscow, Russian Federation. [7]European Molecular Biology Laboratory, Heidelberg, Germany. [8]Dep of Biochemistry and Biophysics and Science for Life Laboratory, Solna, Sweden. [9]Linderstrøm-Lang Centre for Protein Science, Department of Biology, University of Copenhagen, Copenhagen, Denmark. [10]Department of Biochemistry and Biophysics University of California, San Francisco, CA, USA. [11]Department of Structural Cell Biology, Max Planck Institute of Biochemistry, Martinsried, Germany. [12]Université de Montpellier, Centre de Recherche en Biologie Cellulaire de Montpellier (CRBM) CNRS, Montpellier, France. [13]Faculty of Arts and Sciences, Division of Science, Harvard University, Cambridge, MA, USA. [14]Biozentrum, University of Basel, Basel, Switzerland. [15]School of Chemistry and Molecular Biology, University of Queensland, Brisbane, Queensland, Australia. [16]Institute of Cancer Research, London, UK. [17]Cambridge Institute for Medical Research, Department of Haematology, The University of Cambridge, Cambridge, UK. [18]Institute of Molecular Systems Biology, ETH Zürich, Zürich, Switzerland. [19]These authors contributed equally: Mehmet Akdel, Douglas E V Pires, Eduard Porta Pardo, Jürgen Jänes, Arthur O Zalevsky, Bálint Mészáros, Patrick Bryant, Lydia L. Good, Roman A Laskowski. ✉e-mail: alfonso.valencia@bsc.es; so@fas.harvard.edu; janani.durairaj@gmail.com; d.ascher@uq.edu.au; thornton@ebi.ac.uk; norman.davey@icr.ac.uk; amelie.stein@bio.ku.dk; arne@bioinfo.se; tic20@cam.ac.uk; pbeltrao@ethz.ch

## Methods

### Coverage comparison between the SWISS-MODEL repository and AlphaFold2 databases

The SMR and AF2 databases were accessed on 24 July 2021. Reference proteomes for 11 species shared between AF2 and SMR were downloaded from the Uniprot release 2021_03. Only structures corresponding to entries from the reference proteomes were used for the analysis. Numpy[58], Pandas[59], Prody[60] and Matplotlib[61] Python libraries were used for the analysis and the visualization. Structure counters for protein domains were extracted from the corresponding InterPro entries[62]. Code and data are available online (https://github.com/aozalevsky/alphafold2_vs_swissmodel/).

### Comparison between human RoseTTAFold Pfam domain and AlphaFold2 structures

We used the 17,006 human proteins that were defined as the principal isoform for their corresponding gene according to APPRIS and whose sequences were the same in ENSEMBL and Uniprot. We used Pfamscan to identify PFAM domains in the 17,006 protein sequences. The database of PFAM-A models was downloaded on 29 June 2021 and created on 19 March 2021. We kept only those PFAM domains identified with an e-value below $1 \times 10^{-8}$. AF2 models for human proteins were downloaded on 23 July 2021 from https://alphafold.ebi.ac.uk. We extracted the sequences and compared them with the ENSEMBL protein sequences used for the structural analysis. For comparison purposes, all the analyses and results presented here are based on the subset of 17,006 protein sequences for which the ENSEMBL and Alpha-Fold protein sequences were identical. We also extracted pLDDT values for each residue from the AlphaFold models, as these are stored as if they were the B-factor of the protein coordinates file. The RoseTTAFold models were downloaded from the EBI website on 27 July 2021, and the r.m.s.d. between models from both methods was calculated using the function struct.aln from the R package bio3d. All statistical analyses were done using R 4.0.2. Graphical plots were created with the packages ggplot2 (ref. [63]), patchwork and reshape2. Molecular graphics and analyses performed with UCSF Chimera[64], developed by the Resource for Biocomputing, Visualization, and Informatics at the University of California, San Francisco.

### Disorder prediction

Benchmarking data for ordered and disordered protein regions were taken from the benchmark set of IUPred2 (ref. [22]) and were filtered for proteins for which AF2-predicted structures are available in the Alpha-Fold database. Relative SASA was calculated by determining the absolute SASA using DSSP and then comparing it to the SASA calculated in a GGXGG conformation. Receiver operating characteristic (ROC) curves plotting the true positive rate as the function of the false positive rate were calculated on a per-residue level. Area under the ROC curves are single-number measures of the overall predictive performance in the range of 0.5 (for random predictions) to 1.0 (for perfect predictions).

### Exploration of structural space covered by the AlphaFold database compared with the Protein Data Bank

We use the 365,198 proteins from the current AlphaFold database (AF) and 104,323 proteins from PDB in 2016 (until CASP12) with a 100% sequence identity threshold, removing duplicates. Owing to the presence of low-confidence regions in the AF proteins, we first performed trimming to split each AF protein into smaller high-confidence units as follows: a one-dimensional Gaussian filter with a standard deviation $\sigma$ of 5 is applied to the sequence of pLDDT scores extracted from the Cα atoms. The resulting scores are used to split the protein into continuous segments of residues with smoothed pLDDT scores > 70. Segments with fewer than 50 residues are discarded. This removed 68,890 AF proteins with too few high-confidence residues for accurate structural comparison.

For each AF protein segment, and for each PDB protein, rotation invariant moments O3, O4, O5 and F were calculated for the Cα atoms using a k-mer-based approach (with $k = 16$) and radius-based approach ($r = 10$ Å) using Geometricus[24]. These were then converted into shape-mers using a resolution of four for the k-mer based approach and six for the radius-based approach. Shape-mers were counted across the whole protein for a PDB protein and across all splits for an AF protein to give the shape-mer count vectors. We then created a term frequency inverse document frequency (TFIDF) matrix for all PDB and AF proteins, in which the terms are shape-mers and each protein is equivalent to a document. We performed topic modeling using NMF, which attempts to factorize a matrix of size $n \times m$ into matrices $W$ of size $n \times p$, and $H$ of size $p \times m$. We interpret this as finding $p$ topics (here set to 250), each of which consists of a weighted combination of the $m$ shape-mers (defined by $H$). Each of the $n$ proteins can then be seen as a weighted combination of these $p$ topics (defined by $W$).

For topic analysis, we assigned proteins to each topic using knee detection with a weight cut-off[65]. For visualization, we performed t-SNE dimensionality reduction on the $W$ matrix returned by NMF. Topic-specific scores were obtained for each residue within a shape-mer by multiplying the corresponding topic weight for the shape-mer (from $H$) with an RBF kernel score of the Euclidean distance between the residue and the central residue of the shape-mer. These were aggregated across all shape-mers within a protein to obtain the topic-specific residue scores for the protein.

Code and scripts can be found at: https://github.com/TurtleTools/alphafold-structural-space

### Structure-based variant effect predictions using experimental and predicted structures

A subset of experimentally characterized mutations was curated from ThermoMutDB[27], comprising 2,648 single point missense mutations across 132 unique globular proteins. The experimentally measured effect of the mutations on protein stability was represented as the difference in $\Delta\Delta G$ (in kcal/mol) between wild type and mutant. Experimental structures were obtained from the PDB[1]. Homology models were generated using Modeller[66] using the most complete available template within each identity threshold range (20% ± 5%, 30% ± 5%, until 90% ± 5%). AF2 models were generated locally. These mutations were analyzed by computational predictive tools, including the sequence-based predictors I-Mutant[67], SAAFEC-SEQ[68] and MUpro[69], and the structure-based predictors mCSM-stability[70], DUET[71], SDM[72], DynaMut[73], MAESTRO[74], ENCoM[75], DynaMut2 (ref. [33]) and FoldX[31]. For each method, the performance and concordance between the experimental and predicted $\Delta\Delta G$ are determined and presented in the corresponding figures as Pearson's correlation values. A larger set of experimentally determined impact of missense mutations was derived from a compilation of Deep Mutational Scanning (DMS) experiments[29] comprising 117,135 total mutations in 33 proteins. These were compared against predictions made with DynaMut2, FoldX and Rosetta[30,32] (see also Supplementary Methods).

### Pocket and structural motif prediction

We downloaded structures from the AlphaFold Protein Structure Database[76] except for analyses of the LBSp dataset[37]. In the latter case, we used locally modeled structures, as many LBSp structures are from species not included in the public database. We detected pockets and calculated overlap metrics (F-score, Matthew's correlation coefficient (MCC)) using AutoSite[77] from ADFRsuite version 1.0, and OpenBabel[78] was used to prepare PDBQT input for AutoSite (obabel -h -xr–partialcharge gasteiger). For enzyme activity predictions, we selected AlphaFold models without corresponding entries in SWISS-MODEL 2021–11–30, and kept 921 structures with mean pLDDT ≥ 70 and 100–500 residues. We considered 170 proteins as having known enzymatic activity if there was an EC number and/or a catalytic activity annotation in the corresponding UniProt records.

## Oligomerization state prediction

To test the ability of AF2 to predict the oligomeric state of homo-oligomeric assemblies, we downloaded the dataset from Ponstingl et al.[42]. Since the PDB files were not provided, the dataset was filtered to entries for which the oligomeric state was in agreement with PISA annotation. Because AF2's training was done only on single chains, we reasoned that examples, even if they overlap with the training set, could be used to evaluate AF2's oligomeric state prediction capabilities. For each PDB entry, the sequence of chain A was extracted, and a multiple sequence alignment was generated using the automated MMseqs2 webserver through ColabFold. For homo-oligomeric prediction, each MSA was copied, padded with gaps to the total length reflecting the number of copies in the assembly and concatenated. These concatenated alignments were fed into AF2. No templates were used. All five ptm-fine tuned model parameters were used. To test the robustness of AF2's five model parameters to predict homo-oligomeric structures, we use the worst of the predicted TMscores for each state.

## Fold-and-dock prediction of heterodimeric protein complexes

We used 219 heterodimeric complexes from Dockground benchmark 4 (ref. [79]). This set contains unbound forms of heterodimeric protein chains, which share at least 97% sequence identity with the bound forms. The dataset consists of 54% eukaryotic proteins, 38% bacterial proteins and 8% from mixed kingdoms, for example one bacterial protein interacting with one eukaryotic protein. To evaluate performance, one model for each pair was generated with AF2 (using default parameters, except that model_2 was used, providing a complementary set of results to those derived in ref. [20]). To enable docking, we changed only the residue number so that both chains are treated as a long chain with a 200-residue gap, as in ref. [80]. We compared AF2 predictions with models docked with GRAMM. The GRAMM models were ranked using the AACE18 scoring function[81]. Docking quality was estimated with DockQ[45].

## Complex structure predictions for disordered proteins

Predictions were run using the sequences defined in the PDB files (not including modified residues and other molecules). Predictions were done using the Google Colab notebooks by S. Ovchinnikov; homooligomers were predicted using the notebook accessible at https://colab.research.google.com/github/sokrypton/ColabFold/blob/main/AlphaFold2.ipynb, and heterooligomers were predicted using the dedicated notebook, accessible at https://colab.research.google.com/github/sokrypton/ColabFold/blob/main/AlphaFold2_complexes.ipynb. In the case of dimers, the default settings were used. In case of higher order oligomers, one chain was used on its own (usually the IDR if there is only one), and the rest of the chains were concatenated using a long linker (either several 'U's or several repeats of 'SG's).

## Evaluation of AlphaFold2 models for use in experimental model building

AF2 models were used as an aid to rebuilding the existing 6O85 in ISOLDE, with a preliminary implementation pLDDT-based weighting of its existing adaptive distance restraints[46]. Initial fetching and alignment of the relevant AF2 models for each chain used a tool available in pre-release versions of ChimeraX 1.3, allowing command-based fetching of predictions from the AlphaFold EBI server by UniProt ID. For existing models fetched from the wwPDB, the UniProt ID for each chain is automatically parsed from the mmCIF metadata, and each fetched prediction is aligned and renamed to match the target chain. ISOLDE's reference-model distance restraint scheme has four adjustable parameters controlling the restraint potential: kappa (the overall strength); wellHalfWidth (the range over which the restraining force is linearly related to distance); tolerance (the width of a flat-bottom—that is, zero-force—region close to the target); and fallOff (the rate at which the potential tapers at large distances). With the exception of kappa, each of these terms is expressed as a function of the reference interatomic distance: for a given restraint, the final harmonic well width, tolerance and fall-off all increase with increasing reference distance. For the purpose of this study, we added terms to further adjust kappa, tolerance and fallOff according to the pLDDT of the lowest-confidence atom in each restrained pair; all restraints where at least one reference atom had a pLDDT < 50 were disabled. For each chain in the complex, the working model was restrained against the AlphaFold reference using the 'isolde restrain distances' command with the above modifications enabled but otherwise standard settings. Backbone and side chain torsions were also restrained against the reference model using the 'isolde restrain torsions' command with default arguments. After energy minimization and equilibration, the model was inspected and, where necessary, interactively remodeled; where reference model restraints clearly disagreed with the model, they were selectively released. Where the AF2 models included previously-unmodeled residues supported by the density, they were merged into the working model. The final model was refined with phenix.real_space_refine[52] using settings defined by the 'isolde write phenixRsr' command.

For the recapitulation of 7OGG, the AF2 predictions for its two chains were fetched in ChimeraX, as above. The rigid core of each chain was extracted using a community clustering approach based on the PAE matrix; source code for this is available at https://github.com/tristanic/pae_to_domains. After setting B-factors to a constant value of 50, these were used to generate a fresh molecular replacement (MR) solution using PHASER[51]. The original, complete AF2 models were aligned to the MR result in ChimeraX, and occupancies for atoms that were not part of the MR models were set to zero. The result was used as the starting model for rebuilding in ISOLDE. After it was initially settled into the map with distance and torsion restraints applied, the model was inspected and rebuilt end-to-end. During this initial rebuilding, zero-occupancy atoms with clear correspondence to density were reinstated to full occupancy, while residues with no associated density were deleted. Where there was clear disagreement with the map (primarily at the heterodimer interface), the initial distance and torsion restraints were selectively released in favor of interactive remodeling. The resulting model was refined with phenix.refine[52], using settings defined by the 'isolde write phenixRefine' command. In a second and third round of interactive rebuilding in ISOLDE (during which the distance and torsion restraints were fully released) interspersed with phenix.refine, a small number of residues deleted in the first step were re-introduced.

In both the above cases, the final coordinates have been shared with the original authors.

## Reporting summary

Further information on research design is available in the Nature Research Reporting Summary linked to this article.

## Data availability

Contiguous protein regions of human high-confidence structural predictions with no previous structural predictions by homology models in the SWISS-MODEL Repository are available in Supplementary Table 1 and in Github: https://github.com/aozalevsky/alphafold2_vs_swissmodel. The benchmark dataset used for testing of disorder predictions metrics is available in Supplementary Table 2, and predicted disordered regions for human proteins are available in Supplementary Dataset 1 and are integrated into ProViz22 at http://slim.icr.ac.uk/projects/alphafold?page=alphafold_proviz_homepage. The grouping of proteins by structure similarly using the NMF analysis of structural fragments is available as Supplementary Dataset 2, and the pocket prediction scores for 912 human proteins with no previous experimental or predicted structural models are available in Supplementary Table 3.

## Code availability

Coverage comparison between SMD and AF2: https://github.com/aozalevsky/alphafold2_vs_swissmodel/. Exploration of structural space: https://github.com/TurtleTools/alphafold-structural-space.

Pocket predictions: https://github.com/jurgjn/af2_pockets. Protein complexes: https://gitlab.com/ElofssonLab/FoldDock, https://colab.research.google.com/github/sokrypton/ColabFold/blob/main/AlphaFold2.ipynb. Model building: https://github.com/tristanic/pae_to_domains.

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

## Acknowledgements

B. M. has received funding from the European Union's Horizon 2020 research and innovation programme under the Marie Skłodowska-Curie grant agreement no. 842490 (MIMIC). A. Stein received funding from the Lundbeck Foundation (R272-2017-4528) and Novo Nordisk Foundation (NNF18OC0033950). D. B. A. received funding from the National Health and Medical Research Council of Australia (GNT1174405) and the Victorian Government's Operational Infrastructure Support Program. K. L.-L. received funding from the Novo Nordisk Foundation (NNF18OC0033950). D. E. V. P. received funding from an Oracle Research Grant. A. O. Z. received funding from the Russian Science Foundation (RSF) 20-14-00121. A. E., A. Shenoy, G. P., P. Bryant and W. Z. received funding from the Swedish Research Council for Natural Science, grant No. VR-2016-06301, the Knut and Alice Wallenber foundation, the Swedish National Initiative for computing, grant No SNIC 2021/6-197 and Berzelius-2021-29, and Swedish E-science Research Center. T. I. C. received funding from the Wellcome Trust. E. P. P. has received funding from PID2019-107043RA-I00 and RYC2019-026415-I from the Spanish Science Ministry. A. V. has been supported by RTI2018-096653-B-I00. S. O. is supported by NIH Grants DP5OD026389 and R21AI156595, NSF Grant MCB2032259 and the Simons Foundation 735929LPI. P. B. is supported by the Helmut Horten Stiftung and the ETH Zurich Foundation.

## Author contributions

E. P. P., A. V. and V. R. S. performed the AF2 and RoseTTaFold comparison. A. O. Z. performed the AF2 and SMD comparison. J. D., M. A., A. B. and A. V. K. performed the characterization of structural elements in the AlphaFold database. N. E. D. and B. M. performed the disorder region and disordered complex structure prediction. T. I. C. performed the experimental model building studies with assistance from J. B. and A. F. S. O. performed the homo-oligomeric state predictions. J. J. and P. Beltrao did the pocket prediction analysis. A. E., A. Shenoy, G. P., P. Bryant, W. Z. and P. K. performed the protein-protein interaction modelling. J. M. T., N. B. and R. A. L. performed the template search study. A. S. D., P. Beltrao, A. Stein, K. L.-L., L. L. G., C. H. M. R., D. B. A. and D. E. V. P. performed the variant effect prediction analyses. D.B provided technical assistance with running AF2 predictions. A. F. and S. V. contributed suggestions for the analysis and revised the text. P. Beltrao, A. V., A. Stein, K. L.-L., N. E. D., T. I. C., S. O., A. E., J. M. T., J. D. and D. B. A. supervised the work. All authors contributed to the writing of the manuscript.

## Competing interests

The authors declare no competing interests.

## Additional information

**Correspondence and requests for materials** should be addressed to Alfonso Valencia, Sergey Ovchinnikov, Janani Durairaj, David B. Ascher, Janet M. Thornton, Norman E. Davey, Amelie Stein, Arne Elofsson, Tristan I. Croll or Pedro Beltrao.

# Article

# Reporting Summary

## Statistics

For all statistical analyses, confirm that the following items are present in the figure legend, table legend, main text, or Methods section.

| n/a | Confirmed | |
|---|---|---|
| ☐ | ☒ | The exact sample size (*n*) for each experimental group/condition, given as a discrete number and unit of measurement |
| ☒ | ☐ | A statement on whether measurements were taken from distinct samples or whether the same sample was measured repeatedly |
| ☒ | ☐ | The statistical test(s) used AND whether they are one- or two-sided<br>*Only common tests should be described solely by name; describe more complex techniques in the Methods section.* |
| ☒ | ☐ | A description of all covariates tested |
| ☒ | ☐ | A description of any assumptions or corrections, such as tests of normality and adjustment for multiple comparisons |
| ☐ | ☒ | A full description of the statistical parameters including central tendency (e.g. means) or other basic estimates (e.g. regression coefficient) AND variation (e.g. standard deviation) or associated estimates of uncertainty (e.g. confidence intervals) |
| ☒ | ☐ | For null hypothesis testing, the test statistic (e.g. *F*, *t*, *r*) with confidence intervals, effect sizes, degrees of freedom and *P* value noted<br>*Give P values as exact values whenever suitable.* |
| ☒ | ☐ | For Bayesian analysis, information on the choice of priors and Markov chain Monte Carlo settings |
| ☒ | ☐ | For hierarchical and complex designs, identification of the appropriate level for tests and full reporting of outcomes |
| ☐ | ☒ | Estimates of effect sizes (e.g. Cohen's *d*, Pearson's *r*), indicating how they were calculated |

*Our web collection on statistics for biologists contains articles on many of the points above.*

## Software and code

Policy information about availability of computer code

| Data collection | for the analysis and the visualization we used: Numpy 1.23, Pandas 1.4.3, Prody 2.0, and Matplotlib 3.5, R 4.0.2, ggplot2, I-Mutant, SAAFEC-SEQ, MUpro, mCSM-stability, DUET, SDM, DynaMut, MAESTRO, ENCoM, DynaMut2, FoldX, Rosetta.<br>Our own code can be found at<br>Coverage comparison between SWISS-MODEL Repository and AlphaFold2  https://github.com/aozalevsky/alphafold2_vs_swissmodel/<br>Exploration of structural space https://github.com/TurtleTools/alphafold-structural-space<br>Pocket predictions https://github.com/jurgjn/af2_pockets<br>Protein complexes https://gitlab.com/ElofssonLab/FoldDock<br>https://colab.research.google.com/github/sokrypton/ColabFold/blob/main/AlphaFold2.ipynb<br>Model building https://github.com/tristanic/pae_to_domains |
|---|---|
| Data analysis | for the analysis and the visualization we used: Numpy 1.23, Pandas 1.4.3, Prody 2.0, and Matplotlib 3.5, R 4.0.2, ggplot2, I-Mutant, SAAFEC-SEQ, MUpro, mCSM-stability, DUET, SDM, DynaMut, MAESTRO, ENCoM, DynaMut2, FoldX, Rosetta.<br>Our own code can be found at<br>Coverage comparison between SWISS-MODEL Repository and AlphaFold2  https://github.com/aozalevsky/alphafold2_vs_swissmodel/<br>Exploration of structural space https://github.com/TurtleTools/alphafold-structural-space<br>Pocket predictions https://github.com/jurgjn/af2_pockets<br>Protein complexes https://gitlab.com/ElofssonLab/FoldDock<br>https://colab.research.google.com/github/sokrypton/ColabFold/blob/main/AlphaFold2.ipynb<br>Model building https://github.com/tristanic/pae_to_domains |

For manuscripts utilizing custom algorithms or software that are central to the research but not yet described in published literature, software must be made available to editors and reviewers. We strongly encourage code deposition in a community repository (e.g. GitHub). See the Nature Portfolio guidelines for submitting code & software for further information.

## Data

Policy information about availability of data

All manuscripts must include a data availability statement. This statement should provide the following information, where applicable:

- Accession codes, unique identifiers, or web links for publicly available datasets
- A description of any restrictions on data availability
- For clinical datasets or third party data, please ensure that the statement adheres to our policy

The SWISS-MODEL repository (https://swissmodel.expasy.org/repository) and AlphaFold2 (https://alphafold.ebi.ac.uk/) databases were accessed on 24.07.2021. Contiguous protein regions of human high confidence structural predictions with no previous structural predictions by homology models in the SWISS-MODEL Repository are available in Supplementary Table 1 and in Github https://github.com/aozalevsky/alphafold2_vs_swissmodel. The benchmark dataset used for testing of disorder predictions metrics is available in Supplementary Table 2 and predicted disordered regions for human proteins is available as Supplementary dataset 1 and integrated into ProViz22 at http://slim.icr.ac.uk/projects/alphafold?page=alphafold_proviz_homepage. The grouping of proteins by structure similarly using the non-negative Matrix Factorization analysis of structural fragments is available as Supplementary Dataset 2 and the pocket prediction scores for 912 human proteins with no previous experimental or predicted structural models is available in Supplementary Table 3.

## Human research participants

Policy information about studies involving human research participants and Sex and Gender in Research.

| | |
|---|---|
| Reporting on sex and gender | *Use the terms sex (biological attribute) and gender (shaped by social and cultural circumstances) carefully in order to avoid confusing both terms. Indicate if findings apply to only one sex or gender; describe whether sex and gender were considered in study design whether sex and/or gender was determined based on self-reporting or assigned and methods used. Provide in the source data disaggregated sex and gender data where this information has been collected, and consent has been obtained for sharing of individual-level data; provide overall numbers in this Reporting Summary. Please state if this information has not been collected. Report sex- and gender-based analyses where performed, justify reasons for lack of sex- and gender-based analysis.* |
| Population characteristics | *Describe the covariate-relevant population characteristics of the human research participants (e.g. age, genotypic information, past and current diagnosis and treatment categories). If you filled out the behavioural & social sciences study design questions and have nothing to add here, write "See above."* |
| Recruitment | *Describe how participants were recruited. Outline any potential self-selection bias or other biases that may be present and how these are likely to impact results.* |
| Ethics oversight | *Identify the organization(s) that approved the study protocol.* |

Note that full information on the approval of the study protocol must also be provided in the manuscript.

# Field-specific reporting

Please select the one below that is the best fit for your research. If you are not sure, read the appropriate sections before making your selection.

☒ Life sciences          ☐ Behavioural & social sciences          ☐ Ecological, evolutionary & environmental sciences

For a reference copy of the document with all sections, see nature.com/documents/nr-reporting-summary-flat.pdf

# Life sciences study design

All studies must disclose on these points even when the disclosure is negative.

| | |
|---|---|
| Sample size | Samples sizes were determined as the data points available for each computational analysis. All statistical tests considered the sample size for the determination of significance. |
| Data exclusions | No data was excluded from the analysis |
| Replication | All attempts at replication were successful |
| Randomization | Protein structures, regions and variants were randomly allocated for analysis |
| Blinding | The investigators were blind to group allocations during data analysis |

# Reporting for specific materials, systems and methods

We require information from authors about some types of materials, experimental systems and methods used in many studies. Here, indicate whether each material, system or method listed is relevant to your study. If you are not sure if a list item applies to your research, read the appropriate section before selecting a response.

## Materials & experimental systems

| n/a | Involved in the study |
|-----|----------------------|
| ☒ ☐ | Antibodies |
| ☒ ☐ | Eukaryotic cell lines |
| ☒ ☐ | Palaeontology and archaeology |
| ☒ ☐ | Animals and other organisms |
| ☒ ☐ | Clinical data |
| ☒ ☐ | Dual use research of concern |

## Methods

| n/a | Involved in the study |
|-----|----------------------|
| ☒ ☐ | ChIP-seq |
| ☒ ☐ | Flow cytometry |
| ☒ ☐ | MRI-based neuroimaging |

