## [Peer Review File. · Nature Structural & Molecular Biology]

Peer Review Information

Journal: Nature Structural and Molecular Biology

Manuscript Title: A structural biology community assessment of AlphaFold2 applications

Corresponding author name(s): Dr Pedro Beltrao

Editorial Notes:

Reviewer Comments & Decisions:

Decision Letter, initial version:

13th Dec 2021

Dear Dr. Beltrao,

Thank you again for submitting your manuscript "A structural biology community assessment of AlphaFold 2 applications". We now have comments (below) from the 3 reviewers who evaluated your paper. In light of those reports, we remain interested in your study and would like to see your response to the comments of the referees, in the form of a revised manuscript.

You will see that a specific concern was whether the structures used in the analyses had been a part of the AlphaFold2 training set, and a requirement to discuss the performance of the AlphaFold2 beyond the 'known'. Please be sure to address/respond to all concerns of the referees in full in a point-by-point response and highlight all changes in the revised manuscript text file. If you have comments that are intended for editors only, please include those in a separate cover letter.

We expect to see your revised manuscript within 6 weeks. If you cannot send it within this time, please contact us to discuss an extension; we would still consider your revision, provided that no similar work has been accepted for publication at NSMB or published elsewhere.

Reporting Summary:

When submitting the revised version of your manuscript, please pay close attention to our [href="https://www.nature.com/nature-research/editorial-policies/image-integrity">Digital Image Integrity Guidelines. and to the following points below:](https://www.nature.com/nature-research/editorial-policies/image-integrity)

Please note that all key data shown in the main figures as cropped gels or blots should be presented in

uncropped form, with molecular weight markers. These data can be aggregated into a single supplementary figure item. While these data can be displayed in a relatively informal style, they must refer back to the relevant figures. These data should be submitted with the final revision, as source data, prior to acceptance, but you may want to start putting it together at this point.

Data availability: this journal strongly supports public availability of data. All data used in accepted papers should be available via a public data repository, or alternatively, as Supplementary Information. If data can only be shared on request, please explain why in your Data Availability Statement, and also in the correspondence with your editor. Please note that for some data types, deposition in a public repository is mandatory - more information on our data deposition policies and available repositories can be found below:

<https://www.nature.com/nature-research/editorial-policies/reporting-standards#availability-of-data>

[Redacted]

Sincerely,

Sara Osman, Ph.D.
Associate Editor
Nature Structural & Molecular Biology

Referee expertise:

Referee #1: Computational structural biology

Referee #2: Structural method development

Referee #3: Structural bioinformatics

Reviewers' Comments:

Reviewer #1:

Remarks to the Author:

In the present submission "A structural biology community assessment of AlphaFold 2 applications" M. Akdel et al. investigate the recent AlphaFold2 software package across a large variety of current structure prediction challenges.

The field of structural biology has held the long-loved dream of protein structure prediction based on sequence. In the last decade, first co-evolution based approaches have led to significant progress by providing access to contact map predictions, which can complement structure prediction as spatial constraints.

Going beyond contact map prediction, the last 5 years have resulted in approaches based on deep learning. Last year, Alpha Fold 2 has "entered the fray" with claims having solved structure prediction in the presence of sufficient sequence information by mining the vast sequence and structural protein databases via multiple complementary deep learning techniques. This claim was strongly corroborated by AF2 success in Caps14, a blind prediction challenge.

In the present submission "A structural biology community assessment of AlphaFold 2 applications" M. Akdel et al. investigates the suitability of the recent AlphaFold2 software package across a large variety of current structure prediction challenges.

The field of structural biology has held the long-loved dream of protein structure prediction based on sequence. In the last decade, first co-evolution based approaches have led to significant progress by providing access to contact map predictions, which can complement structure prediction as spatial constraints. Going beyond contact map or, later, distance map prediction, the last 5 years have resulted in approaches based on deep learning. Last year, Alpha Fold 2 has "entered the fray" with claims having solved structure prediction in the presence of sufficient sequence information by mining the vast sequence and structural protein databases via multiple complementary deep learning techniques. This claim was strongly corroborated by AF2 success in Casp14, a blind prediction challenge.

The present submission evaluates this new framework and assesses its progress compared to other

state-of-the art tools. The different presented tasks are well suited to highlight AF2's performance. There are some remarkable results, for example the prediction of oligomeric assemblies. The resented work is thus an extremely timely, important and comprehensive external look on of the most important contributions to scientific progress in the last year. All relevant statistics and uncertainties are reported. The conclusions are robust, well-validated and justified and I agree with the authors final statement that AF2 "will have a transformative impact in life sciences." There is credit to prior work (but this could be expanded). The introduction in to the field, the comparison to other methods, the actual presentation and quality of writing is very high. The SI presents relevant details. After very minor revision, I would recommend publication of the present submission.

Major Issues:

-

Minor issues:

- Introduction: The last decade has strongly transformed the field, e.g. after the introduction of DCA. The authors focus in the introduction on AF2, but "it's standing on the shoulders of giants". Both raw structural and sequence data and algorithms/ mathematical frameworks have been considerably expanded and I would therefore recommend expanding the introduction.
- Fig 2.: How was this figure generated, what are the axis? Are the axis from a clustering algorithm? Is so, which one?
- Fig.3: The font in the figure is very small. The authors should test, whether a larger font is possible.
- p. 15 "The remarkable accuracy of AF2 predictions even in the face of little to no sequence similarity to existing structures provides clear opportunities for future use in experimental model building". Can the authors speculate as to why AF2 performs so well even if not trained on this family?
- P. 18 "In summary, we find that AF2 models, when considering their uncertainty, can be applied to existing structural biology challenges with near experimental quality." It is my understanding that AF2 still relies on the presence of a MSA with a high number of effective sequences? One of the "holy grails" in structural biology is structure prediction from a single sequence alone so I would add this restriction. What are possible other current limitations the authors still see?
- Some refs. are incomplete (Mirdita et al., Pozzati et al.).

Reviewer #2:

Remarks to the Author:

Akdel et al. describe applications of AlphaFold2 ranging from missense variants, function- and ligand binding, protein interactions and experimental structure modeling. The manuscript is a community effort to show and evaluate the possibilities of AlphaFold2 prediction in downstream analyses.

My take home message of this manuscript is that if the predicted structure has a high pLDDT (>90) then the downstream tasks perform similarly well compared to experimental structures. The work is very valuable for the community to understand how to apply AlphaFold2 results. While I am overall excited about this work there are some experimental details missing that would require clarification or additional experiments. Additionally, the reproducibility of the study should be improved.

Major:

- For all experiments it is unclear if the underlying structures were part of the training of AlphaFold2. If yes, I expect to rerun the analyses with structures that were not seen during training.
- Complex prediction was already benchmarked in multiple preprints (Mirdita et al., Bryant et al. and

Evans et al.). However, it is not mentioned in the introduction. I assume this was not mentioned because of the fast pace of development in the field. However, since the manuscript was submitted a month after these predictors were released, I would expect to see these in the introduction.

- For some of the experiments the code is available, however this is not the case for all. Please add the missing analysis scripts. Additionally, the code is distributed over many Github repositories, some are missing a license file and the links are mentioned at multiple locations in the manuscript. It would be great to have a single central point for all scripts (perhaps as a "code availability section").

- Fig. 5S claims that these topics (NMF clusters) without PDB representatives are rare new folds. Did you try to find SCOP/CATH domains in these structures? Some of these structures seem to also be multi-domain proteins. I am mainly familiar with the term fold in the context of single domains.

Minor:

- For variant prediction tools like Rosetta and FoldX were used to compute the effect of mutations. As a reader I would find it interesting to discuss the SNP effect on the predicted structure. Do some mutations result in strong structural changes?

- Is the template identity in Fig. 3D really causal for the drop in performance or is it due to template coverage? I assume that a template covering the full sequence would still result in a good model.

- "Thus, application of AF2 may have allowed us to answer the question of what is the shortest length of repeat that forms a beta-solenoid". Does this mean there is no shorter repeat possible than this? Is this a physical limitation?

- I find the analysis using shape-mers and a NMF elegant to dissect the structural space. How much does the k-mer amino acid space overlap with the structural space?

- At multiple locations in the manuscript AlphaFold2 is named AlphaFold. I would recommend to consistently name it to avoid confusions.

- Multiple citations are missing journals or preprint servers:

"Limits and potential of combined folding and docking using PconsDock."

"ColabFold - Making protein folding accessible to all"

"Predicting and interpreting large scale mutagenesis data using analyses of protein stability and conservation."

"Improved prediction of protein-protein interactions using AlphaFold2 and extended multiple-sequence alignments."

- Citations (Pozzati et al.) and (Mirdita et al.) in the main manuscript are missing the year.

- "community have indicated that AF2 can predict the structure of complexes, which it was not initially trained to handle." <- citation missing.

- Reading the manuscript shows me how excited the authors are about AlphaFold2s results. However, I would recommend from refraining from using "remarkable" too often.

- For reviewing line numbers would help.

Reviewer #3:

Remarks to the Author:

Given the advent of AF2 there have been a slew of paper in the literature and preprints looking at aspects of its potential use. These papers have varied in their quality.

This paper offers a proper perspective across many of the areas where AF2 is likely to be used as an important tool. It is well written and gives a clear motivation as to the need for the analysis that is carried out. The paper is also timely as it helps to place correctly how AF2 can and should be used and also how and where further developments are needed.

First section of the extra amount of the proteome that can now be accurately predicted is clear and the analysis sound.

Section 2 on IDRs shows as other studies have also commented that AF2 (in this case combining its SASA and per residue confidence scores) out performed state of the art intrinsic disorder predictors.
- It would be useful to note in the discussion of the results on IDRs that these are primarily defined as regions that are not solved by x-ray crystallography – therefore directly correlate to the way AF2 was trained. There is some discussion as to whether this definition of IDRs is the best one – but it is certainly the one with the most available experimental data. So this is more to note why AF2 may be so good here.

Performed a well-constructed search for rare structural motifs and domains in the high confidence af2 predictions. The results demonstrate that these are found and offer some interesting examples of the biology that could be extracted from them.

- I was unclear if there were any examples of entirely novel structural motifs or domains from AF2 – this is important to consider if it is able to extrapolate beyond the known (or if we believe all motifs have been sampled already in the PDB)

The authors also demonstrate that AF2 structures with confident predictions can be used to generate structural hypotheses about the potential impact of disease or trait-associated mutations.

- In this section I was not entirely clear what how af2 was being used to predict the impact of a mutation to be correlated with what experimental data – Greater clarity in the results part of the main text would be useful here

The section of pockets was clear and well described - clearly identifying the caveats that pLDDT of predicted pockets though useful to know when to use AF2 pockets may be biased as they are more likely to have similar templates already available.

Complex building is an area that is rapidly moving in relation to ML methods and AF2 - the section in the paper describes the promise that is evident here.

AF2 use for experimental model building.

It would be interesting to discuss a little more about how much difference this is likely to make particularly noting the comments that even high confidence residues from AF2 can be incorrect and manual inspection is still needed to correct these.

Author Rebuttal to Initial comments

Please find below our point-by-point responses to the reviewer's concerns. Our response is highlighted in blue. To facilitate the evaluation of the revision we have also highlighted in the manuscript in blue the major changes made in response to the reviewer's concerns.

Reviewers' Comments:

Reviewer #1:

Remarks to the Author:

In the present submission "A structural biology community assessment of AlphaFold 2 applications" M. Akdel et al. investigate the recent AlphaFold2 software package across a large variety of current structure prediction challenges.

The field of structural biology has held the long-loved dream of protein structure prediction based on sequence. In the last decade, first co-evolution based approaches have led to significant progress by providing access to contact map predictions, which can complement structure prediction as spatial constraints.

Going beyond contact map prediction, the last 5 years have resulted in approaches based on deep learning. Last year, Alpha Fold 2 has "entered the fray" with claims having solved structure prediction in the presence of sufficient sequence information by mining the vast sequence and structural protein databases via multiple complementary deep learning techniques. This claim was strongly corroborated by AF2 success in Caps14, a blind prediction challenge.

In the present submission "A structural biology community assessment of AlphaFold 2 applications" M. Akdel et al. investigates the suitability of the recent AlphaFold2 software package across a large variety of current structure prediction challenges.

The present submission evaluates this new framework and assesses its progress compared to other state-of-the art tools. The different presented tasks are well suited to highlight AF2's performance. There are some remarkable results, for example the prediction of oligomeric assemblies. The presented work is thus an extremely timely, important and comprehensive external look on of the most important contributions to scientific progress in the last year. All relevant statistics and uncertainties are reported. The conclusions are robust, well-validated and justified and I agree with the authors final statement that AF2 "will have a transformative impact in life sciences." There is credit to prior work (but this could be expanded). The introduction in to the field, the comparison to other methods, the actual presentation and quality of writing is very high. The SI presents relevant details. After very minor revision, I would recommend publication of the present submission.

We thank the reviewer for the positive assessment of the work.

Major Issues:

-

Minor issues:

-Introduction: The last decade has strongly transformed the field, e.g. after the introduction of DCA. The authors focus in the introduction on AF2, but “it’s standing on the shoulders of giants”. Both raw structural and sequence data and algorithms/ mathematical frameworks have been considerably expanded and I would therefore recommend expanding the introduction.

We certainly agree with the reviewer that these recent developments in prediction of protein structures rely on progress that has been made over many years, including the availability of experimentally solved structures for training. It makes perfect sense to expand the introduction to cover this material. At the same time it is difficult to do justice to this past work in a way that is fair for all scientists involved. The recent review that we cite in the introduction mentions over 50 citations that describe the progression of methodologies and datasets that have led to the developments in the methods as they exist today. Nevertheless, we tried to expand the introduction to better reflect the past developments that led up to AF2.

-Fig 2.: How was this figure generated, what are the axis? Are the axis from a clustering algorithm? Is so, which one?

The figure is the visualisation of a t-SNE dimensionality reduction analysis and the 2 axis are units of the lower dimension projection (t-SNE dimension 1 and t-SNE dimension 2). These are analogous to the first 2 components in the principal component analysis. However, the relationship between distance on the plot between two points and their similarity is highly non-linear and therefore the units in the axis are best thought of as “arbitrary” units. The t-SNE dimensionality reduction is only used for visualisation purposes. We have clarified in the figure legend what the visualisation is.

-Fig.3: The font in the figure is very small. The authors should test, whether a larger font is possible.

We have increased the font sizes as suggested.

-p. 15 “The remarkable accuracy of AF2 predictions even in the face of little to no sequence similarity to existing structures provides clear opportunities for future use in experimental model building”. Can the authors speculate as to why AF2 performs so well even if not trained on this family?

We believe the reviewer is referring to the prediction of the Nse5/6 complex structure that follows from this sentence. In this case, there are other structures of this family but with low sequence identity to this ones modelled here. Alphafold2’s capacity to predict structures that are very difficult targets has been to a large extent benchmark by others through the CASP competition. Even in the CASP benchmarks for difficult targets it is possible to say that there aren’t any proteins in the training set with identical sequences but it will not be straightforward to exclude that there aren’t proteins in the training set with at least partially similar folds. Speculating, it is quite possible that AlphaFold2 has learned to generalise the folding “rules” from the training set to potential new folds. However, we also believe this is an aspect that remains to be fully evaluated. We have expanded the discussion section and now include some discussion about the degree of generalisation that AlphaFold2 makes.

-P. 18 “In summary, we find that AF2 models, when considering their uncertainty, can be applied to existing structural biology challenges with near experimental quality.” It is my understanding that AF2 still relies on the presence of a MSA with a high number of effective sequences? One of the “holy grails” in structural biology is structure prediction from a single sequence alone so I would add this restriction. What are possible other current limitations the authors still see?

We very much agree with the reviewer that AF2 still is limited to cases where the MSA carries enough information to allow for the predictions. There is work from Mohammed AlQuraishi’s lab attempting to make predictions without the need for MSA (<https://doi.org/10.1101/2021.08.02.454840>) but this also shows a lower performance. Other limitations that are so far apparent are that AF2 is not appropriate to predict structures of mutated proteins (<https://doi.org/10.1101/2021.09.19.460937>, <https://doi.org/10.1038/s41594-021-00714-2>), not well suited for the understanding of the potential diversity of conformations that a protein might have, and of course it is also not trained to predict the structure of protein interactions with other biomolecules such as DNA/RNA/metabolites. Finally, it is also unclear the extent by which AF2 has explicitly learned aspects of protein biophysics and if such information can be provided to the end user. We have now included a short discussion about such limitations in the manuscript.

-Some refs. are incomplete (Mirdita et al., Pozzati et al.).

We thank the reviewer and have corrected this.

Reviewer #2:

Remarks to the Author:

Akdel et al. describe applications of AlphaFold2 ranging from missense variants, function- and ligand binding, protein interactions and experimental structure modeling. The manuscript is a community effort to show and evaluate the possibilities of AlphaFold2 prediction in downstream analyses.

My take home message of this manuscript is that if the predicted structure has a high pLDDT (>90) then the downstream tasks perform similarly well compared to experimental structures. The work is very valuable for the community to understand how to apply AlphaFold2 results. While I am overall excited about this work there are some experimental details missing that would require clarification or additional experiments. Additionally, the reproducibility of the study should be improved.

Major:

- For all experiments it is unclear if the underlying structures were part of the training of AlphaFold2. If yes, I expect to rerun the analyses with structures that were not seen during training.

We appreciate that it is important to evaluate the applications of AlphaFold2 while taking care to focus on proteins that were not included in the training set. There are some results that were derived from proteins or protein regions that were included in the training set, such as part of the

results with the VEPs (Fig 3A, 3D) and the pocket analysis (Fig4A-C). However, in all cases, we show the performance on protein regions for which there are no experimental structural models, and therefore could have not have been included in the training set. This includes the a section of the performance of the variant effect predictions (Fig 3B); the prediction of enzyme function (Fig 4C, Table S3); all cases of protein complex predictions since complexes were not used for training (Fig 5); and the use of AlphaFold2 for structural data refinement of the Nse5/6 complex (Fig 6F,G,H). We believe that these analyses on proteins or protein regions without experimental models meet the requirements set out in this concern.

To further expand the analyses of proteins that had no experimental structures that could have been used for training we have revised the analyses where we attempt to identify pockets and predict enzyme function (revised Fig 4C). In this analysis we now use 910 human proteins that have no experimental structure and no homology model in the SWISS-MODEL repository. We show in this revised analysis that AlphaFold2 models can be used to predict enzyme function with reasonable accuracy.

- Complex prediction was already benchmarked in multiple preprints (Mirdita et al., Bryant et al. and Evans et al.). However, it is not mentioned in the introduction. I assume this was not mentioned because of the fast pace of development in the field. However, since the manuscript was submitted a month after these predictors were released, I would expect to see these in the introduction.

We have now updated the introduction to cover these and other recent results from what is a fast moving area of research.

- For some of the experiments the code is available, however this is not the case for all. Please add the missing analysis scripts. Additionally, the code is distributed over many Github repositories, some are missing a license file and the links are mentioned at multiple locations in the manuscript. It would great to have a single central point for all scripts (perhaps as a "code availability section").

We have added additional analysis scripts to GitHub, and created a table summarising code availability that is in the new code availability section as suggested by the reviewer.

- Fig. 5S claims that these topics (NMF clusters) without PDB representatives are rare new folds. Did you try to find SCOP/CATH domains in these structures? Some of these structures seem to also be multi-domain proteins. I am mainly familiar with the term fold in the context of single domains.

We agree with the reviewer that the analysis on protein topics may need to be clarified. We have tried to introduce in the results section what these protein groups represent - proteins having similar combinations of structural elements. It is important to be clear that while some combinations of structural elements might equate to folds, other combinations of structural elements may simply reflect even the presence of specific binding sites or other combinations of structural motifs. Another important caveat is the degree of resolution of clustering of proteins into groupings based on these combinations of structural elements. As with every clustering approach, one needs to select a similarity cut-off to define the groups, and this decision will

impact on the identification of groups with or without any PDB representation. In regards to the the topics in Fig 5S, we meant to say that these groups represent cases where there were very few PDB structures. As they all contain at least some PDB structures they do not represent new combinations of structural elements or novel folds. We have revised the statement to indicate that these groups of proteins have combinations of structural elements that are not very well covered by PDB structures. It is possible that, varying the clustering parameters, we may identify smaller groupings that would truly correspond to novel arrangements of structural elements and/or novel folds. However, we think this may take additional considerable effort that may be more suited for a follow up study.

Minor:

- For variant prediction tools like Rosetta and FoldX were used to compute the effect of mutations. As a reader I would find it interesting to discuss the SNP effect on the predicted structure. Do some mutations result in strong structural changes?

If we understood this concern correctly, the reviewer is curious to know what would be the AlphaFold2 predicted structures for some of the missense mutations of known impact that were studied using Rosetta and FoldX. However, we don't think that AlphaFold2 is particularly well suited to predict the structures of mutated proteins. There have been a few attempts to do so as reported in the literature so far (<https://doi.org/10.1101/2021.09.19.460937>, <https://doi.org/10.1038/s41594-021-00714-2>) and some of our own fairly small attempts so far also indicate that, at least for small changes like single point mutations, AlphaFold2 is not appropriate to study such structural changes. We have added this limitation as areas of further development in the discussion section.

- Is the template identity in Fig. 3D really causal for the drop in performance or is it due to template coverage? I assume that a template covering the full sequence would still result in a good model.

The mutations tested were always on regions where the homologous structure had coverage to build the homology model. In addition, we only see minimal difference in coverage at different identity thresholds (see the table below). Based on this we believe the decrease in performance is specifically tied to the lower sequence identity of the template and not a change in coverage.

identity cut-off	Average coverage
20	86.48
30	88.13
40	91.00
50	92.76
60	94.51

70	96.48
80	95.96
90	89.82

- "Thus, application of AF2 may have allowed us to answer the question of what is the shortest length of repeat that forms a beta-solenoid". Does this mean there is no shorter repeat possible than this? Is this a physical limitation?

We primarily meant to say that up until this finding, it would currently be the shortest length of repeat that forms a beta-solenoid. We revised the description of this to avoid the suggestion that no shorter repeat is possible.

- I find the analysis using shape-mers and a NMF elegant to dissect the structural space. How much does the k-mer amino acid space overlap with the structural space?

The analysis aims to look at the differences in structural space covered by AlphaFold DB and PDB, which is why we did not make any attempt to look at sequence space. Using amino acid k-mer counts is not as straightforward due to the sparsity that explodes with comparable k-mer sizes as what we used for the shape-mers, and due to the fact that some amino acids are more similar than others which would not be taken into account. The shape-mer approach, though using a nomenclature similar to "k-mer", is not the same concept as the discretization can be controlled by the resolution threshold. Thus, a comparison between sequence and structural spaces would need more thought and is beyond the scope of this analysis.

- At multiple locations in the manuscript AlphaFold2 is named AlphaFold. I would recommend to consistently name it to avoid confusions.

We agree with the reviewer and for consistency we now use AlphaFold2 or AF2 throughout the manuscript.

- Multiple citations are missing journals or preprint servers:

"Limits and potential of combined folding and docking using PconsDock."

"ColabFold - Making protein folding accessible to all"

"Predicting and interpreting large scale mutagenesis data using analyses of protein stability and conservation."

"Improved prediction of protein-protein interactions using AlphaFold2 and extended multiple-sequence alignments."

We have added the suggested references and tried to improve the introduction or discussion section to better reflect the fast moving pace of this research.

- Citations (Pozzati et al.) and (Mirdita et al.) in the main manuscript are missing the year.

We have corrected the references.

- "community have indicated that AF2 can predict the structure of complexes, which it was not initially trained to handle." <- citation missing.

We have added the references as suggested.

- Reading the manuscript shows me how excited the authors are about AlphaFold2s results. However, I would recommend from refraining from using "remarkable" too often.

We agree and have toned down these types of subjective remarks.

- For reviewing line numbers would help.

We have added numbers and apologise for not providing them in a previous version.

Reviewer #3:

Remarks to the Author:

Given the advent of AF2 there have been a slew of paper in the literature and preprints looking at aspects of its potential use. These papers have varied in their quality.

This paper offers a proper perspective across many of the areas where AF2 is likely to be used as an important too. It is well written and gives a clear motivation as to the need for the analysis that is carried out. The paper is also timely as it helps to place correctly how AF2 can and should be used and also how and where further developments are needed.

We thank the reviewer for the positive remarks.

First section of the extra amount of the proteome that can now be accurately predicted is clear and the analysis sound.

Section 2 on IDRs shows as other studies have also commented that AF2 (in this case combining its SASA and per residue confidence scores) out performed state of the art intrinsic disorder predictors.

- It would be useful to note in the discussion of the results on IDRs that these are primarily defined as regions that are not solved by x-ray crystallography – therefore directly correlate to the way AF2 was trained. There is some discussion as to whether this definition of IDRs is the best one – but it is certainly the one with the most available experimental data. So this is more to note why AF2 may be so good here.

We agree with the reviewer and have added a point about this in the discussion.

Performed a well-constructed search for rare structural motifs and domains in the high confidence af2 predictions. The results demonstrate that these are found and offer some interesting examples of the biology that could be extracted from them.

- I was unclear if there were any examples of entirely novel structural motifs or domains from AF2 – this is important to consider if it is able to extrapolate beyond the known (or if we believe all motifs have been sampled already in the PDB)

To be clear, we did not find any topic/cluster that did not have any PDB structures within them but we did find some that only had a very small number of representatives. However, the definition of a topic depends on parameters that can be tuned and as any clustering exercise

one may find more or less specific groupings of larger or smaller size. So it is still quite likely that there will be proteins with combinations of structural motifs that are truly novel. We think that a follow up study will be needed to fully explore the space of structural motifs present in the growing compilation of AlphaFold2 predicted structures. We do think that this initial exploration showcases already how valuable this will be. We have revised the manuscript to make these results clearer and to discuss also the future potential expansions.

The authors also demonstrate that AF2 structures with confident predictions can be used to generate structural hypotheses about the potential impact of disease or trait-associated mutations.

- In this section I was not entirely clear what how af2 was being used to predict the impact of a mutation to be correlated with what experimental data – Greater clarity in the results part of the main text would be useful here

We appreciate that there may be some confusion in regards to this section. AlphaFold2 was used to predict the structures and the impact of mutations were predicted with other tools that can use structures (predicted or experimental) to predict the impact of mutations. A different potential approach could have been to use AlphaFold2 to predict the structure of the reference and mutated proteins and somehow have compared these structures to evaluate the impact of the mutations. However, as discussed in response to the second reviewer, we don't think AlphaFold2 is appropriate to predict the structures of mutated proteins. We now make this distinction in the discussion section which will hopefully also help clarify what was done in this section.

The section of pockets was clear and well described - clearly identifying the caveats that pLDDT of predicted pockets though useful to know when to use AF2 pockets may be biased as it they are more likely to have similar templates already available.

Complex building is an area that is rapidly moving in relation to ML methods and AF2 - the section in the paper describes the promise that is evident here.

We appreciate the positive comments in regards to these sections of the work. We have also improved the introduction/discussion sections to account for recent progress that has been made in these areas.

AF2 use for experimental model building.

It would be interesting to discuss a little more about how much difference this is likely to make particularly noting the comments that even high confidence residues from AF2 can be incorrect and manual inspection is still needed to correct these.

We agree that the use of AlphaFold2 predictions for experimental modelling won't really replace manual inspection. However, we note these examples provided were also meant to emphasise that AlphaFold2 does make some mistakes in high confidence regions. For the majority of high confidence regions the AlphaFold2 predictions were correct. We have expanded the discussion around this topic to make the point that AlphaFold2 will not do away with experimental studies and the combination of experimental data collection plus artificial intelligence is likely to be the growing trend of application.

Decision Letter, first revision:

Our ref: NSMB-A45589A

5th Apr 2022

Dear Dr. Beltrao,

Thank you for submitting your revised manuscript "A structural biology community assessment of AlphaFold2 applications" (NSMB-A45589A). It has now been seen by the original referees and their comments are below. The reviewers find that the paper has improved in revision, and therefore we'll be happy in principle to publish it in Nature Structural & Molecular Biology, pending minor revisions to satisfy the referees' final requests and to comply with our editorial and formatting guidelines.

To facilitate our work at this stage, we would appreciate if you could send us the main text as a word file. Please make sure to copy the NSMB account (cc'ed above).

Sincerely,
Sara

Sara Osman, Ph.D.
Associate Editor
Nature Structural & Molecular Biology

Reviewer #1 (Remarks to the Author):

The authors have sufficiently addressed my concerns and comments. I would recommend publication of the revised submission.

Reviewer #2 (Remarks to the Author):

I would like to thank the authors for addressing my comments.

Minor:

- Citations (Pozzati et al.) and (Mirdita et al.) in the main manuscript are missing the year.

The citation for Pozzati et al. is now correct but Mirdita et al. is still missing the year.

Reviewer #3 (Remarks to the Author):

The authors have comprehensively answered my comments and updated the manuscript and I have no more comments to add

Decision Letter, final checks:

Our ref: NSMB-A45589A

30th Jun 2022

Dear Dr. Beltrao,

Thank you for your patience as we've prepared the guidelines for final submission of your Nature Structural & Molecular Biology manuscript, "A structural biology community assessment of AlphaFold2 applications" (NSMB-A45589A). Please carefully follow the step-by-step instructions provided in the attached file, and add a response in each row of the table to indicate the changes that you have made. Please also check and comment on any additional marked-up edits we have proposed within the text. Ensuring that each point is addressed will help to ensure that your revised manuscript can be swiftly handed over to our production team.

In recognition of the time and expertise our reviewers provide to Nature Structural & Molecular Biology's editorial process, we would like to formally acknowledge their contribution to the external peer review of your manuscript entitled "A structural biology community assessment of AlphaFold2 applications". For those reviewers who give their assent, we will be publishing their names alongside the published article.

Nature Structural & Molecular Biology offers a Transparent Peer Review option for new original research manuscripts submitted after December 1st, 2019. As part of this initiative, we encourage our authors to support increased transparency into the peer review process by agreeing to have the reviewer comments, author rebuttal letters, and editorial decision letters published as a Supplementary item. When you submit your final files please clearly state in your cover letter whether

or not you would like to participate in this initiative. Please note that failure to state your preference will result in delays in accepting your manuscript for publication.

Cover suggestions

As you prepare your final files we encourage you to consider whether you have any images or illustrations that may be appropriate for use on the cover of Nature Structural & Molecular Biology.

Nature Structural & Molecular Biology has now transitioned to a unified Rights Collection system which will allow our Author Services team to quickly and easily collect the rights and permissions required to publish your work. Approximately 10 days after your paper is formally accepted, you will receive an email in providing you with a link to complete the grant of rights. If your paper is eligible for Open Access, our Author Services team will also be in touch regarding any additional information that may be required to arrange payment for your article.

Please note that *Nature Structural & Molecular Biology* is a Transformative Journal (TJ). Authors may publish their research with us through the traditional subscription access route or make their paper immediately open access through payment of an article-processing charge (APC). Authors will not be required to make a final decision about access to their article until it has been accepted. [Find out more about Transformative Journals](https://www.springernature.com/gp/open-research/transformative-journals)

Authors may need to take specific actions to achieve [compliance with funder and institutional open access mandates](https://www.springernature.com/gp/open-research/funding/policy-compliance-faqs). If your research is supported by a funder that requires immediate open access (e.g. according to [Plan S principles](https://www.springernature.com/gp/open-research/plan-s-compliance)) then you should select the gold OA route, and we will direct you to the compliant route where possible. For authors selecting the subscription publication route, the journal's standard licensing terms will need to be accepted, including [self-archiving policies](https://www.springernature.com/gp/open-research/policies/journal-policies). Those licensing terms will supersede any other terms that the author or any third party may assert apply to any version of the manuscript.

Please note that you will not receive your proofs until the publishing agreement has been received

through our system.

Please use the following link for uploading these materials:
[Redacted]

Best regards,

Sophia Frank
Editorial Assistant
Nature Structural & Molecular Biology
nsmb@us.nature.com

On behalf of

Sara Osman, Ph.D.
Associate Editor
Nature Structural & Molecular Biology

Reviewer #1:

Remarks to the Author:

The authors have sufficiently addressed my concerns and comments. I would recommend publication of the revised submission.

Reviewer #2:

Remarks to the Author:

I would like to thank the authors for addressing my comments.

Minor:

- Citations (Pozzati et al.) and (Mirdita et al.) in the main manuscript are missing the year.

The citation for Pozzati et al. is now correct but Mirdita et al. is still missing the year.

Reviewer #3:

Remarks to the Author:

The authors have comprehensively answered my comments and updated the manuscript and I have no more comments to add

Final Decision Letter:

20th Sep 2022

Dear Dr. Beltrao,

We are now happy to accept your revised paper "A structural biology community assessment of AlphaFold2 applications" for publication as a Article in Nature Structural & Molecular Biology.

As soon as your article is published, you can generate your shareable link by entering the DOI of your article here: http://authors.springernature.com/share.

Corresponding authors will also receive an automated email with the shareable link

Note the policy of the journal on data deposition:

<http://www.nature.com/authors/policies/availability.html>.

Your paper will be published online soon after we receive proof corrections and will appear in print in the next available issue. You can find out your date of online publication by contacting the production

team shortly after sending your proof corrections. Content is published online weekly on Mondays and Thursdays, and the embargo is set at 16:00 London time (GMT)/11:00 am US Eastern time (EST) on the day of publication. Now is the time to inform your Public Relations or Press Office about your paper, as they might be interested in promoting its publication. This will allow them time to prepare an accurate and satisfactory press release. Include your manuscript tracking number (NSMB-A45589B) and our journal name, which they will need when they contact our press office.

About one week before your paper is published online, we shall be distributing a press release to news organizations worldwide, which may very well include details of your work. We are happy for your institution or funding agency to prepare its own press release, but it must mention the embargo date and Nature Structural & Molecular Biology. If you or your Press Office have any enquiries in the meantime, please contact press@nature.com.

Please note that *Nature Structural & Molecular Biology* is a Transformative Journal (TJ). Authors may publish their research with us through the traditional subscription access route or make their paper immediately open access through payment of an article-processing charge (APC). Authors will not be required to make a final decision about access to their article until it has been accepted. [Find out more about Transformative Journals](https://www.springernature.com/gp/open-research/transformative-journals)

Sincerely,
Sara

Sara Osman, Ph.D.
Associate Editor
Nature Structural & Molecular Biology
